# Syncphony: Synchronized Audio-to-Video Generation with Diffusion Transformers

**Jibin Song**[1,2]**, Mingi Kwon**[1,2]**, Jaeseok Jeong**[1,2]**, Youngjung Uh**[1]

[1]Department of Artificial Intelligence, Yonsei University
[2]CineLingo*

{sjbpsh1,kwonmingi,jete_jeong,yj.uh}@yonsei.ac.kr

## Abstract

Text-to-video and image-to-video generation have made rapid progress in visual quality, but they remain limited in controlling the precise timing of motion. In contrast, audio provides temporal cues aligned with video motion, making it a promising condition for temporally controlled video generation. However, existing audio-to-video (A2V) models struggle with fine-grained synchronization due to indirect conditioning mechanisms or limited temporal modeling capacity. We present *Syncphony*, which generates 380×640 resolution, 24fps videos synchronized with diverse audio inputs. Our approach builds upon a pre-trained video backbone and incorporates two key components to improve synchronization: (1) **Motion-aware Loss**, which emphasizes learning at high-motion regions; (2) **Audio Sync Guidance**, which guides the full model using a visually aligned off-sync model without audio layers to better exploit audio cues at inference while maintaining visual quality. To evaluate synchronization, we propose **CycleSync**, a video-to-audio-based metric that measures the amount of motion cues in the generated video to reconstruct the original audio. Experiments on AVSync15 and The Greatest Hits datasets demonstrate that **Syncphony** outperforms existing methods in both synchronization accuracy and visual quality. Project page is available at: https://jibin86.github.io/syncphony_project_page

## 1 Introduction

Video generation has achieved remarkable progress especially in text-to-video (T2V) and image-to-video (I2V). They synthesize visually crisp and temporally coherent videos that match the given text prompt and/or a starting frame. However, we still need additional ways to control the motions that are difficult to control by the texts or the starting frames. For example, texts inherently lack explicit timings of when and how motions would occur, although they may describe motions, e.g., "dog barking" and "striking bowling". In what rhythm would the dog bark? When is the ball released, how fast does it roll, and when does it hit the pins? Similarly, image-based conditions also face inherent limitations. An image can convey information about the appearance, pose, background, and layout of the scene, but it represents only a static snapshot of a single moment.

In contrast, audio signals inherently carry temporal clues because audio and video share the same temporal axis. Returning to the earlier examples, the accompanying audio would provide how many times and exactly when the dog barks, when the ball is released, how quickly it travels, and when it hits the pins. Therefore, we tackle generating videos that are synchronized to audios.

Even with audio, text, or image conditions, existing audio-to-video methods (Lee et al., 2023; Jeong et al., 2023; Yariv et al., 2023; Zhang et al., 2024) struggle with fine-grained synchronization between audio and motion. These approaches rely on indirect mappings, such as magnitude-based adjustments (Lee et al., 2023) or audio-to-text projections (Jeong et al., 2023; Yariv et al., 2023), which fail to reflect the complex and detailed temporal structures in audio signals. Instead, we directly inject audio features into the visual generation process via a cross-attention mechanism, enabling audio-motion alignment. In parallel, compared to T2V models (Jin et al., 2024a; HaCohen

---

*{jibinsong,mingikwon,jaeseokchung}@cinelingo-labs.com

et al., 2024; Blattmann et al., 2023; Wan et al., 2025) which generate high-resolution, high-frame-rate, temporally coherent videos, Zhang et al. (2024) adds temporal layers to an image backbone, training them from scratch with limited data (e.g., 6 fps at 256×256 resolution) leading to broken temporal coherence, such as flickering and saturation artifacts. We address this by leveraging a pre-trained video backbone with strong temporal modeling capabilities, resulting in more stable and consistent motion.

Despite these advancements, audio-to-video generation still faces a fundamental challenge: MSE-based objectives alone are insufficient for modeling accurate motion timing and appropriate motion magnitude. In diffusion or flow models, MSE is effective for reconstructing overall visual appearance, but it provides weak supervision for localized, precisely timed motion changes. Therefore, stronger and more targeted supervision is needed in regions that exhibit significant motion.

To this end, we propose Syncphony, which generates high-quality videos at 380×640 resolution, 24fps, and up to 5 seconds in length, and most of all, synchronized to audio. We design Syncphony to have joint self-attention of text-video and audio cross-attention with RoPE on top of a DiT architecture.

For training, we introduce a motion-focused loss that places greater emphasis on regions exhibiting significant visual movement. This encourages the model to better detect motion that is causally linked to the audio signal, even when such motion is highly localized. Furthermore, for sampling, we introduce a novel synchronization guidance strategy that enhances audio-driven motion without compromising visual fidelity. Motivated by the observation that applying traditional classifier-free guidance to audio conditions makes it difficult to train scenes without audio, we instead propose a guidance method that skips the audio layer itself, rather than dropping the audio condition.

Also, we provide a comprehensive set of experiments that evaluate synchronization, visual quality, and semantic alignment across real-world scenarios. In particular, we propose a novel synchronization metric, CycleSync, designed for high-frame-rate video generation, overcoming the limitations of existing metrics that assume unrealistic one-to-one audio-video mappings or operate only at low temporal resolution. Using CycleSync, we demonstrate that our approach successfully models varied audio-motion dynamics. Overall, our method, Syncphony, outperforms existing approaches across all aspects, and we will release our code, models, and evaluation tools to support future research in this direction.

## 2 RELATED WORKS

### 2.1 TEXT&IMAGE-TO-VIDEO GENERATION

**Models.** Based on the autoregressive models (Yan et al., 2021; Hong et al., 2022; Jin et al., 2024b) and diffusion models (Ho et al., 2022; Brooks et al., 2024), video generative models have been advanced dramatically. Notably, adapting DiT allows huge improvements in high-quality video generation with scalability (Peebles & Xie, 2023; Chen et al., 2023; Wang et al., 2023; HaCohen et al., 2024). Chen et al. (2024), and Valevski et al. (2024) proposed a hybrid approach that combines autoregressive and diffusion models. Upon them, Jin et al. (2024a) proposes both a spatial and temporal feature compression enabling the generation of long videos with high fidelity at a lower training cost. Notably, they only allow text or an image as conditions. On the other hand, our method takes audio as condition.

**Guidance.** Guidance mechanisms play a crucial role in improving sample quality across generative models. Classifier-Free Guidance (Ho & Salimans, 2022) interpolates between conditional and null-conditional predictions to enhance visual fidelity, but requires models to be explicitly trained with null conditions. Spatiotemporal Skip Guidance (Hyung et al., 2025) constructs a weak model by skipping visually sensitive layers, and interpolates its predictions with those of the full model to improve quality without additional training. However, in T2I and T2V architectures, visual and semantic representations are often deeply entangled, making such selective skipping difficult.

## 2.2 AUDIO-TO-VIDEO GENERATION

Recent works on Audio-to-Video (A2V) generation have explored how to synthesize temporally aligned videos conditioned on audio inputs. Lee et al. (2023) modulates cross-attention weights based on audio amplitude to control video. Although this approach is simple, amplitude alone does not transfer the semantic and temporal structure of audio, resulting in weak fine-grained synchronization. On the other hand, Jeong et al. (2023); Yariv et al. (2023) project audio embeddings into a text embedding space and generate frames using pre-trained text-to-video (T2V) models. This indirect audio-to-motion mapping is a bottleneck in delivering temporal expressiveness and hinders precise alignment between audio cues and motion transitions. AVSyncD (Zhang et al., 2024) injects audio layers into a Stable Diffusion-based text-to-image (T2I) model, but it is limited to the T2I backbone's spatial resolution and suffers from relatively shallow temporal modeling capacity. Although Zhang et al. (2024) further introduces synchronization guidance, this requires additional training and often causes flickering, degrading visual smoothness. While talking head models have shown strong lip-sync performance for speech (Wang et al., 2025), they are limited to facial motion and human voice. We instead focus on non-speech sounds and general visual motion, which could complement lip-sync systems in real-world audio scenarios.

In contrast to these prior approaches, our method builds on the strengths of diffusion transformer-based T2V models to directly incorporate fine-grained temporal audio cues. By leveraging a high-capacity backbone capable of high-resolution, high-frame-rate generation and introducing targeted synchronization guidance and loss-level modifications, our model achieves accurate audio-motion synchronization across diverse domains while preserving visual fidelity.

**Synchronization metrics.** Existing synchronization metrics, such as RelSync (Zhang et al., 2024) and AlignSync (Zhang et al., 2024), require downsampling to 6 fps, which reduces temporal resolution and undermines the evaluation of fine-grained motion. AV-Align (Yariv et al., 2023) assumes a one-to-one correspondence between motion and audio peaks, which fails to generalize to real-world scenarios involving preparatory or residual motion. For example, a hammer moves before the impact sound and stops at the sound. To address these limitations, we propose a new synchronization metric that supports high frame rates and generalizes to real-world audio–motion scenarios.

## 3 SYNCPHONY

### 3.1 OVERVIEW

Our goal is to generate high-quality videos that have motions aligned with audio inputs. We build upon a pretrained autoregressive diffusion transformer (Jin et al., 2024a), which sequentially synthesizes consecutive video chunks by denoising each chunk for given a previous chunk and a text prompt. As shown in Figure 1, our model takes an initial frame, a text prompt, and an audio waveform as input. The initial frame is encoded into a latent $z_0$ using a VAE, which serves as the starting point for generating video latents $\{z_l\}_{l=1}^{L}$. Text features are extracted from pretrained encoders (Raffel et al., 2020; Radford et al., 2021), and audio features $\{a_i\}_{i=0}^{L_{\text{audio}}}$ are obtained from DenseAV (Hamilton et al., 2024) encoder. Each transformer block includes a joint self-attention layer, which attends over the concatenated sequence of text tokens and video latents. To incorporate audio, we insert a cross-attention layer before the joint self-attention layer in the later blocks, allowing each video latent to attend to its aligned audio segment for fine-grained synchronization.

In the following subsections, we propose a motion-aware loss that puts more weight on the regions with large motions for training (3.2), introduce a sampling strategy designed to sample the videos toward better audio-conditional outputs (3.3), and describe additional architectural details (3.4).

### 3.2 MOTION-AWARE LOSS

Conventional video generation models typically use Mean Squared Error (MSE) loss, which measures the pixel or latent-level discrepancy between predicted and ground-truth frames. While MSE is effective for general reconstruction, it treats all spatial and temporal regions equally, without distinguishing between static and dynamic areas. As a result, even when the model produces inaccurate motion timing or insufficient movement, e.g., a delayed or insufficient gun firing motion, the error

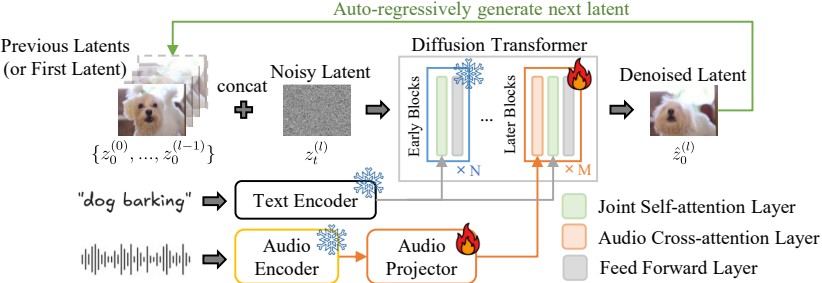

Figure 1: **Overview of our video generation framework.** Given an initial frame, a text prompt, and an audio waveform, the model autoregressively predicts each video latent through iterative denoising. The Diffusion Transformer is divided into two groups of transformer blocks: the early blocks (frozen, blue) and the later blocks (trainable, orange). Text features are injected into all blocks via joint self-attention, whereas audio cross-attention layers are inserted *only into the later (trainable) blocks*. For brevity, latents are visualized as RGB frames, but they are spatiotemporal features extracted by VAE.

remains low if the overall appearance is visually close to the ground truth. This may lead the model to interpret poorly synchronized predictions as successful outputs, weakening its ability to learn precise audio-visual alignment.

This limitation is particularly critical in real-world scenarios where audio cues correspond to distinct, temporally localized motion, such as a drum hit or bowling pin collision. In such cases, accurate timing and appropriate motion magnitude are essential for maintaining natural synchronization. Therefore, it is necessary to provide stronger and more focused supervision to areas involved in high-motion events.

In Figure 2, we observe that latent differences between adjacent frames tend to correlate with audio events, even when the corresponding motion is not clearly visible in the video frames, as in (c). Based on this observation, we propose a **Motion-aware Loss** that amplifies the learning signal according to the intensity of ground-truth motion. This amplifies supervision at moments of significant movement, encouraging the model to better capture and align motion with audio cues.

The proposed loss function is defined as:

$$\mathcal{L} = \mathcal{L}_{\text{base}} + \lambda\,\mathcal{L}_{\text{motion}}, \tag{1}$$

$$\mathcal{L}_{\text{base}} = \|\hat{\epsilon}_t - \epsilon_t^{GT}\|^2, \tag{2}$$

$$\mathcal{L}_{\text{motion}} = \sum_{l=2}^{L} \|(\hat{\epsilon}_t^{(l)} - \epsilon_t^{GT(l)}) \odot \underbrace{(z_{\text{clean}}^{GT(l)} - z_{\text{clean}}^{GT(l-1)})}_{\text{motion}}\|^2. \tag{3}$$

The proposed loss function is defined as:

$$\mathcal{L} = \|\hat{\epsilon}_t - \epsilon_t^{GT}\|^2 + \lambda \sum_{l=2}^{L} \|(\hat{\epsilon}_t^{(l)} - \epsilon_t^{GT(l)}) \odot \underbrace{(z_{\text{clean}}^{GT(l)} - z_{\text{clean}}^{GT(l-1)})}_{\text{motion}}\|^2, \tag{4}$$

where $\hat{\epsilon}_t^{(l)}$ and $\epsilon_t^{GT(l)}$ denote the predicted and ground-truth noise scores at denoising timestep $t$ for the $l$-th temporal position in the video latent sequence, respectively, $z_{\text{clean}}^{GT(l)}$ denotes the ground-truth *clean* latent at position $l$, and $\odot$ denotes element-wise multiplication. The second term weights prediction errors according to the magnitude of ground-truth motion between consecutive frames, with $\lambda$ as a hyperparameter (we set $\lambda = 1$).

(a) striking bowling

(b) dog barking

(c) machine gun shooting

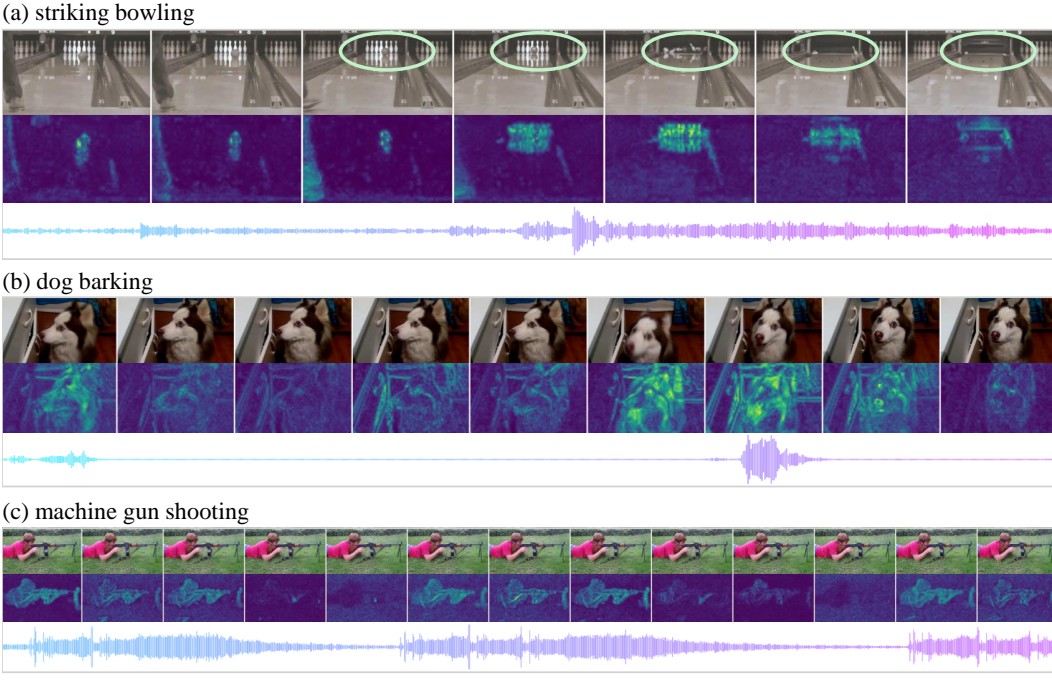

Figure 2: Visualization of video frames (top), latent difference maps (middle), and audio waveforms (bottom) over time. In (a) and (b), the latent differences correspond to key audio events such as pin collisions and barking. In (c), although motion is not clearly visible in raw frames, latent differences still reveal temporal alignment with machine gun audio signals.

This design ensures that prediction errors during dynamic motion are penalized more heavily than those in static periods, encouraging the model to better capture the timing and intensity of important motions.

Importantly, we do not directly use audio signal strength as a supervision signal. This is because audio and motion do not always exhibit a one-to-one temporal alignment: motion may precede or follow audio events, or span multiple frames. For instance, a lion may move before roaring, or a bowling ball may roll before impact. By focusing on ground-truth motion magnitude rather than audio signal strength, our loss design allows the model to learn natural synchronization patterns without rigidly assuming direct temporal alignment.

One might worry that the motion-aware loss could be negatively affected by camera motion or background movement that is unrelated to audio events. However, because our formulation weights the loss based on ground-truth motion intensity itself, the model naturally learns to differentiate between motion that is causally linked to audio cues and motion that is not.

Overall, Motion-aware Loss strengthens the model's attention to motion-relevant regions, encouraging the model to learn diverse audio-motion relationships and generate natural, well-aligned motion sequences. Additional notes on motion-aware loss are provided in Appendix A.

### 3.3 AUDIO SYNC GUIDANCE

In audio-conditioned video generation, audio-driven layers are responsible for injecting timing cues into the visual dynamics. However, these cues from audio are not always strong or clear, so it's hard for the model to determine whether to reflect them in the generated motion. For example, when a drumstick hits a plastic surface, a subtle crinkling sound helps specify the exact target. Relying only on the coarse impact sound can misplace the strike.

To address this, we propose **Audio Sync Guidance** (ASG) that reinforces the influence of audio signals so the model better captures and reflects them in motion. As illustrated in Figure 3(b), we run

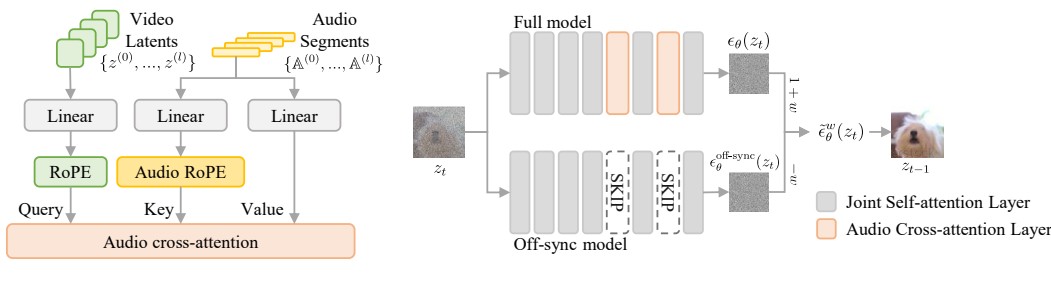

Figure 3: (a) Audio cross-attention with Audio RoPE. Each video latent attends to a local audio segment using cross-attention. RoPE is applied to both video queries and audio keys, using a shared positional embedder to align modalities in relative position space. (b) Audio Sync Guidance. An off-sync model that skips the audio cross-attention layers guides the full model to better utilize audio cues during sampling.

two branches that share the same visual backbone: a *full* model with audio cross-attention layers enabled, and an *off-sync* model where only those layers are disabled. We found that the off-sync model produces outputs that are visually similar to the full model's, yet desynchronized (Please see the supporting experiments in Appendix B.2). Thus, the difference between the two predictions isolates the synchronization component and could serve as guidance for synchronization. By adding this difference back into the full model's output, ASG amplifies the influence of audio and encourages more synchronized motion generation.

Formally, given a latent $z_t$ at denoising timestep $t$, the guided prediction is

$$\tilde{\epsilon}_\theta^w(z_t) = \epsilon_\theta(z_t) + w \left( \epsilon_\theta(z_t) - \epsilon_\theta^{\text{off-sync}}(z_t) \right), \tag{5}$$

where $\epsilon_\theta(z_t)$ is the denoising output of the full model, $\epsilon_\theta^{\text{off-sync}}(z_t)$ is the output with audio layers skipped, and $w$ is the guidance-strength hyperparameter controlling the degree of audio emphasis. For clarity, we omit the integration with Classifier-Free Guidance; please see an Appendix B.3 for the connection to CFG.

In summary, ASG highlights audio cues by disabling only the audio cross-attention layers in the off-sync model, improving audio–motion alignment while preserving visual fidelity without additional training.

### 3.4 ARCHITECTURAL DETAILS

**Training layer selection.** To leverage the pretrained video backbone effectively, we identify which transformer blocks to fine-tune through a layer-wise sensitivity analysis. We find that earlier layers primarily control spatial structure and semantic fidelity, whereas later layers govern temporal dynamics and motion refinement. Based on this, we insert audio-driven cross-attention only into the later blocks and fine-tune them jointly. This strategy allows the model to focus on synchronizing motion with audio signals while maintaining high visual fidelity and leveraging the strong generalization capability of the pretrained I2V backbone. Details are provided in Appendix D.1.2.

**Audio conditioning.** To synchronize video motion precisely with audio cues, we apply Rotary Positional Embedding to inject relative temporal information into the audio features during cross-attention (Audio RoPE), as illustrated in Figure 3(a). We confirm that Audio RoPE leads to tighter temporal alignment between motion and sound events. Implementation details and an ablation study are provided in Appendix D.2.

## 4 EVALUATING AUDIO−MOTION SYNCHRONIZATION

Although prior synchronization metrics (Zhang et al., 2024; Yariv et al., 2023) are useful, they require a low fixed frame-rate or introduce wrong assumption that the peak magnitudes of audio

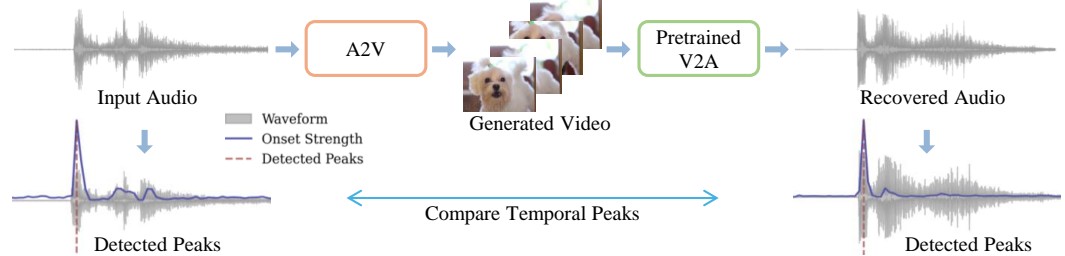

Figure 4: **CycleSync** metric pipeline. The generated video is fed into a pretrained Video-to-Audio (V2A) model to reconstruct audio. We compare temporal peaks between the reconstructed and original audio signals. High peak correspondence indicates that the generated video accurately preserves the timing structure of the original audio.

and video should match. It makes them less reliable for high-frame-rate videos or real-world audio-motion scenarios.

To address these limitations, we propose **CycleSync**, a synchronization metric based on a video-to-audio (V2A) reconstruction process. Instead of directly comparing motion and audio peaks, CycleSync evaluates whether the motion in a video provides enough signal to reconstruct the temporal structure of the original audio. As illustrated in Figure 4, we feed the generated video into a state-of-the-art V2A model (Viertola et al., 2025), and compare the resulting audio to the original input audio by aligning their temporal peaks. By aligning audio peaks between the original and recovered audio, we can assess whether the generated video contains sufficient timing and motion cues to reproduce the original audio structure.

Formally, given an original audio signal $a$ and a generated video $\hat{v}$, we reconstruct the audio $\hat{a}$ using a pretrained video-to-audio model $f_{v2a}$:

$$\hat{a} = f_{v2a}(\hat{v}). \tag{6}$$

Let $\mathbb{A} = P(a)$ and $\hat{\mathbb{A}} = P(\hat{a})$ be the sets of onset peaks extracted from $a$ and $\hat{a}$, respectively. We compute the number of matched peaks under temporal tolerance $\delta$ using one-to-one matching (see Appendix Alg. 1):

$$I = \sum_{a \in \mathbb{A}} \mathbf{1}\left[\exists\, \hat{a} \in \hat{\mathbb{A}},\, |a - \hat{a}| < \delta\right]. \tag{7}$$

The CycleSync score is then computed as an IoU between the two peak sets:

$$\text{CycleSync} = \frac{I}{|\mathbb{A}| + |\hat{\mathbb{A}}| - I}. \tag{8}$$

where $\delta$ is a temporal tolerance and $\mathbf{1}[\cdot]$ is the indicator function.

A higher CycleSync score indicates that the generated video preserves the timing structure of the original audio.

## 5 EXPERIMENT

### 5.1 EXPERIMENTAL SETUP

**Dataset.** We evaluate our model using AVSync15[1] (Zhang et al., 2024) and TheGreatestHits[2] (Owens et al., 2016), whose samples have synchronized audio and video.

---

[1] AVSync15 is a curated subset of the VGGSound dataset consisting of 1,500 videos from 15 action-related classes.

[2] TheGreatestHits is a dataset where a person strikes various objects with drumsticks, producing distinct impact sounds closely tied to visual motion.

Table 1: Quantitative results on the AVSync15 dataset.

| Input | Model | FID ↓ | FVD ↓ | IA ↑ | IT ↑ | CycleSync ↑ | User Study | | |
|---|---|---|---|---|---|---|---|---|---|
| | | | | | | | IQ ↑ | FC ↑ | Sync ↑ |
| T+A | TempoTokens (Yariv et al., 2023) | 8.9 | 4187.2 | 27.24 | 27.88 | $13.10_{\pm1.16}$ | - | - | - |
| I+T | Pyramid Flow (Jin et al., 2024a) | 8.9 | 550.7 | - | 29.34 | $14.25_{\pm1.39}$ | - | - | - |
| | Pyramid Flow (fine-tuned) | **8.5** | 294.6 | - | 30.02 | $12.34_{\pm1.14}$ | - | - | - |
| I+T+A | CoDi () | 14.5 | 1522.6 | 28.15 | 23.42 | | | | |
| | TPoS () | 11.9 | 1227.8 | 38.36 | 30.73 | | | | |
| | AVSyncD (Zhang et al., 2024) | 9.2 | 491.5 | 35.23 | 30.18 | $16.38_{\pm1.38}$ | 30 | 18 | 78 |
| | Ours | **8.5** | **293.1** | **37.02** | **30.23** | $\mathbf{16.48_{\pm1.28}}$ | **270** | **282** | **222** |
| | *Groundtruth* | - | - | 37.06 | 30.18 | $22.15_{\pm1.8}$ | | | |

Table 2: Quantitative results on the TheGreatestHits dataset.

| Input | Model | FID ↓ | FVD ↓ | IA ↑ | IT ↑ | CycleSync ↑ |
|---|---|---|---|---|---|---|
| I+T | Pyramid Flow (Jin et al., 2024a) | **6.5** | 350.5 | - | 18.42 | $7.41_{\pm0.83}$ |
| | Pyramid Flow (fine-tuned) | 6.9 | 195.6 | - | 20.86 | $9.23_{\pm0.92}$ |
| I+T+A | AVSyncD (Zhang et al., 2024) | 6.8 | 327.8 | 12.35 | **21.77** | $9.89_{\pm0.84}$ |
| | Ours | 6.7 | **166.2** | 13.83 | 19.64 | $\mathbf{16.18_{\pm1.26}}$ |
| | *Groundtruth* | - | - | 14.68 | 19.47 | $15.99_{\pm1.5}$ |

**Baselines.** We compare our method against the following baseline models: We employ the Pyramid Flow Video model (**I+T2V**) (Jin et al., 2024a), which conditions on text and image inputs, TempoTokens (**T+A2V**), which conditions on audio and text inputs, and AVSyncD (**I+T+A2V**), which conditions on audio and image inputs. For a closer comparison between I2V and A2V, we also employ a fine-tuned version of our model without audio layers, denoted as Pyramid Flow (fine-tuned).

**Evaluation metrics.** To assess visual quality, we report **FID** (Heusel et al., 2017) (Fréchet Inception Distance) and **FVD** (Unterthiner et al., 2019) (Fréchet Video Distance). FID measures the fidelity of individual frames, while FVD evaluates the spatiotemporal coherence of the entire video. To assess semantic alignment with conditioning modalities, we use **Image-Text Similarity (IT)** and **Image-Audio Similarity (IA)**. IT evaluates how well the generated frames correspond to the input text prompt using CLIP (Radford et al., 2021), while IA measures semantic alignment between audio signals and visual content using ImageBind (Girdhar et al., 2023). To assess audio-motion synchronization, we report **CycleSync**, which evaluates whether the generated videos contain sufficient motion cues synchronized with audio signals. We also conduct a user study on 150 videos from the AVSync15 dataset. Participants compare video pairs across three criteria; **synchronization (Sync)**, **image quality (IQ)**, and **frame consistency (FC)**. Implementation details of the user study are provided in Appendix E.

**Implementation details.** We use the pretrained Pyramid Flow Video model (Jin et al., 2024a) as the backbone. Generated videos are up to 5 seconds long at 24 fps and $380 \times 640$ resolution. Audio is sampled at 16kHz. During training, we randomly sample training clips from different temporal segments of each video to improve generalization to various audio-motion alignments. During evaluation, we extract three 2-second clips at distinct time points per video. The AVSync15 dataset provides 450 clips, and TheGreatestHits provides 732 clips for evaluation. We use CLIP (Radford et al., 2021) and DenseAV (Hamilton et al., 2024) audio backbone as our text encoder and audio encoder, respectively. We train our model on 4 NVIDIA RTX 3090 GPUs (24GB).

## 5.2 MAIN RESULTS

### 5.2.1 MODEL COMPARISON

**Quantitative results.** Tables 1 and 2 show results on AVSync15 and TheGreatestHits. Across both datasets, our model consistently outperforms baselines in synchronization accuracy while maintaining competitive visual and semantic quality. Compared to AVSyncD, our model achieves higher

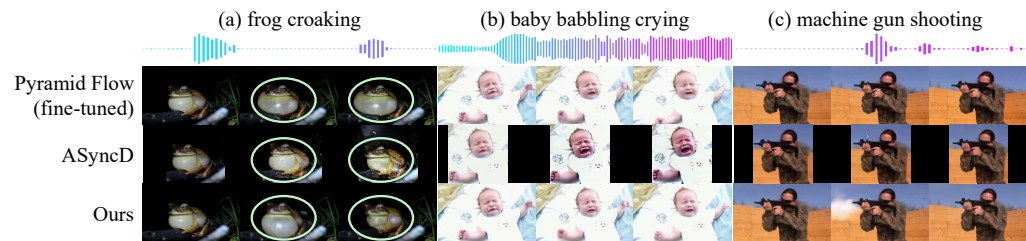

Figure 5: **Qualitative comparison of generated videos.** Our method produces more explicit and temporally consistent motion compared to both baselines.

Table 3: Ablation results on AVSync15.

| Model Variant | FID ↓ | FVD ↓ | CycleSync ↑ |
|---|---|---|---|
| w/o Motion-aware Loss | **8.4** | 305.9 | $15.18_{\pm 1.48}$ |
| Full model w/o ASG | 8.5 | 299.1 | $15.31_{\pm 1.49}$ |
| Full model w/ ASG ($w = 1$) | 8.5 | 294.2 | $15.94_{\pm 1.56}$ |
| Full model w/ ASG ($w = 4$) | 8.7 | 298.3 | $16.26_{\pm 1.4}$ |
| **Full model w/ ASG** ($w = 2$) | 8.5 | **293.1** | $\mathbf{16.48_{\pm 1.28}}$ |

CycleSync scores and lower FID/FVD, indicating improved temporal coherence. User study further confirms these gains, with clear preference for our model in synchronization, image quality, and frame consistency.

On TheGreatestHits, our model even surpasses the ground-truth CycleSync score. We attribute this to the generated videos exhibiting strong and clear motion aligned with audio events, whereas ground-truth videos often contain off-event movements or sounds, such as hovering or background noise. These results suggest that our model demonstrates greater sensitivity to audio cues under synchronization metrics. Additional results using existing metrics (AV-Align, RelSync, AlignSync) are reported in Appendix C.4.

**Qualitative results.** Figure 5 presents qualitative comparisons among ours, Pyramid Flow (fine-tuned), and AVSyncD. Our method produces clearer motion dynamics and stable appearances, whereas AVSyncD often suffers from saturation artifacts and weakened motion. We recommend watching the supplemental videos to see additional qualitative results (Appendix F.1).

### 5.2.2 METRIC COMPARISON

**Controlled metric comparison.** We analyze synchronization robustness under controlled temporal shifts. Details are provided in Appendix C.2. As shown in Figure 7, CycleSync is markedly more sensitive to temporal misalignment than other metrics, clearly differentiating synchronized from desynchronized cases.

**Human alignment validation for CycleSync.** We conduct a user study to assess how well CycleSync aligns with human perception. Details are provided in Appendix C.3. As shown in Table 7 and Table 8, CycleSync achieves the highest positive correlation with human preference, while prior metrics show weak or negative trends.

### 5.3 ABLATION STUDY

**Effect of Motion-aware Loss.** When trained without Motion-aware Loss, the model tends to produce weaker and less clearly timed motions. As shown in Figure 6, it often fails to initiate or terminate motion in sync with the corresponding audio events. Incorporating Motion-aware Loss improves both the magnitude and temporal precision of motion, particularly at the onset and offset of dynamic actions. This is because Motion-aware Loss selectively amplifies learning signals at

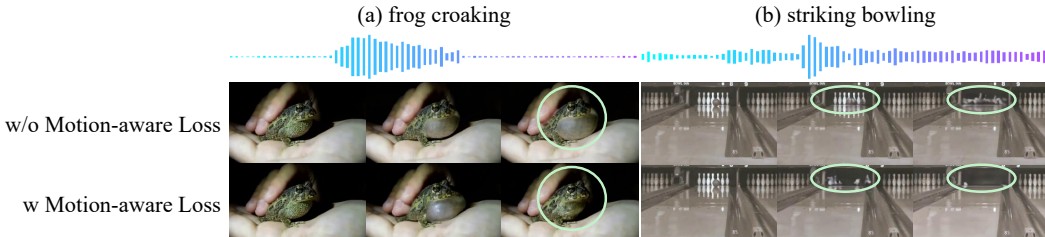

Figure 6: **Ablation of Motion-aware Loss.** (a) Without Motion-aware Loss, the model fails to terminate the motion correctly at the end of the audio. (b) It also fails to trigger motion at the correct audio onset. In contrast, with Motion-aware Loss, the model generates motion that more accurately aligns with the beginning and end of the audio event.

points of high motion intensity, guiding the model to learn more precisely on the timing structure of audio-driven actions.

**Effect of Audio Sync Guidance.** As shown in Tables 3 , applying Audio Sync Guidance (ASG) with scale $w = 2$ improves synchronization metrics while preserving visual fidelity. Increasing the scale to $w = 4$ yields marginal gains in synchronization, but introduces over-exaggerated motion (e.g., frog inflation or recoil motion), which slightly degrades visual realism reflected in higher FVD, while FID remains stable.

## 6 CONCLUSION

We introduced *Syncphony*, a high-quality *audio-synchronized video* generation framework. By conditioning on text, image, and audio inputs, our model captures both the semantic context and the fine-grained temporal dynamics of motion. To improve audio-motion alignment, we incorporated two key techniques: **Motion-aware Loss** encourages accurate timing by emphasizing high-motion regions, and **Audio Sync Guidance** enhances sensitivity to audio signals during inference while maintaining visual quality. To better evaluate synchronization accuracy, we proposed **CycleSync**, a video-to-audio-based metric that measures whether the generated video retains sufficient motion cues to reconstruct the original audio. This enables a more reliable assessment than the existing metrics in real-world scenarios.

**Limitation.** Motion-aware Loss reweights training by ground-truth motion magnitude to emphasize dynamic regions. A current scope of this design is that the weighting is not explicitly selective for audio-relevant motion at the supervision level. This still allows the model to learn audio-motion associations, but a more selective motion proxy (e.g., foreground/action-aware weighting) could further improve robustness in highly dynamic scenes where audio-driven motion should be emphasized. We also note that the limitations of CycleSync as a synchronization metric are discussed in Appendix C.5.

ETHICS STATEMENT

As a generative model, our method could be used to facilitate deceptive interactions that would cause harm, such as fraud. It could be used to impersonate public figures and influence political processes, or as a tool to promote hate speech or abuse. To address this, we will include explicit license terms and usage guidelines to promote ethical and lawful use, referencing best practices such as the Adobe Generative AI User Guidelines. If the model is released, implement safeguards such as prompt or image filtering to restrict high-risk applications, including impersonation or politically manipulative content.

REPRODUCIBILITY STATEMENT

Key components of our implementation are provided in the supplementary materials, and detailed descriptions of our method, training, inference, and evaluation are included in the appendix. We will release our code, trained models, and evaluation tools to ensure reproducibility.

ACKNOWLEDGMENTS

This work was supported by IITP grants [RS-2024-00439762, Developing Techniques for Analyzing and Assessing Vulnerabilities, and Tools for Confidentiality Evaluation in Generative AI Models] and [RS-2020-II201361, Artificial Intelligence Graduate School Program (Yonsei University)] funded by the Korean government (MSIT).

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

# Syncphony: Synchronized Audio-to-Video Generation with Diffusion Transformers

Supplementary Material

## A  ADDITIONAL NOTES ON MOTION-AWARE LOSS

Motion-aware Loss is designed based on ground-truth motion magnitude, not audio amplitude. This reflects the fact that motion peak and audio peak do not always exhibit one-to-one temporal alignment. Instead, audio events are often accompanied by diverse motion patterns that vary by context.

For example, some events, such as a gunshot or a dog's bark, occur with motion and sound nearly perfectly aligned. However, many others do not. A lion may start moving its mouth and body before emitting a roar. A person winds up before throwing a ball. A hammering motion may start before the sound and end just as the impact occurs. A trombone player moves the instrument before the sound begins. A bowling ball rolls with a low rumble before producing a sharp impact sound when it hits the pins.

Therefore, synchronizing motion to audio does not mean matching peak amplitudes. Rather, it involves capturing the causal and contextual patterns of motion that correspond to different types of audio events. Our loss focuses on motion regions, encouraging the model to learn this alignment without relying on rigid audio-based timing. This design encourages the model to learn various audio-motion relationships, leading to natural audio-visual aligned video generation.

## B  AUDIO SYNC GUIDANCE

### B.1  CLASSIFIER FREE GUIDANCE FOR AUDIO GUIDANCE

Conventional classifier-free guidance (CFG) is trained by randomly dropping conditioning inputs, a strategy that has proven highly effective in models such as text-to-image and class-conditional image generation. However, we observe that directly applying this random-drop strategy to audio conditioning leads to degraded performance.

Unlike text, audio carries a meaningful semantic interpretation even when its value is *zero*. In other words, *silence* is itself an informative condition. When audio inputs are randomly dropped during training, the model becomes unable to properly model silence and fails to understand its distinct role compared to the absence of conditioning. Our experiments show that this mismatch significantly harms synchronization quality.

To address this issue, instead of dropping audio conditions, we propose an Audio Sync Guidance mechanism that selectively skips audio-conditioning layers during inference. This approach preserves the semantic meaning of silence while preventing the model from being exposed to ambiguous training signals. In the following section, we describe this strategy in more detail.

### B.2  DIFFERENCES BETWEEN FULL AND OFF-SYNC MODEL IN AUDIO SYNC GUIDANCE

To better understand how Audio Sync Guidance contributes to synchronization, we evaluate whether an off-sync model, formed by skipping the audio layers, can still retain appearance and overall motion. As shown in the last row of Figure 5 and "Off-sync model" of Table 4, the model remains out of synchronization but still preserves appearance (FID) and motion quality (FVD). Since the visual quality remains similar between the full and off-sync models, their difference primarily captures audio-related motion cues. By adding this difference back into the full model's output, Audio Sync Guidance amplifies the influence of audio and encourages more synchronized motion generation.

Table 4: **Analysis of Audio Sync Guidance.** The full model includes audio layers, whereas the off-sync model skips them.

| Model Variant | FID ↓ | FVD ↓ | CycleSync ↑ |
|---|---|---|---|
| Off-sync model | 8.5 | 294.6 | $12.34_{\pm 1.14}$ |
| Full model | 8.5 | 299.1 | $15.31_{\pm 1.49}$ |
| **Full model w/ ASG** | **8.5** | **293.1** | $\mathbf{16.48_{\pm 1.28}}$ |

## B.3 INTEGRATION OF CFG AND AUDIO SYNC GUIDANCE

Classifier-Free Guidance (Ho & Salimans, 2022) extrapolates between conditional (full) and null-conditional predictions to enhance visual fidelity:

$$\tilde{\epsilon}_\theta(z_t) = \epsilon_\theta(z_t, c_\varnothing) + w_t \left( \epsilon_\theta(z_t, c) - \epsilon_\theta(z_t, c_\varnothing) \right) \tag{9}$$

At inference time, the Audio Sync Guidance and CFG are combined additively:

$$\tilde{\epsilon}_\theta(z_t) = \epsilon_\theta(z_t, c) + w_a \left( \epsilon_\theta(z_t, c) - \epsilon_\theta^{\text{off-sync}}(z_t, c) \right) + w_t \left( \epsilon_\theta(z_t, c) - \epsilon_\theta(z_t, c_\varnothing) \right) \tag{10}$$

We take the conditional prediction as the base and add CFG and ASG residuals. In our implementation, we use $w_a = 2$ and $w_t = 4$.

## B.4 AUDIO SYNC GUIDANCE COMPARED TO PRIOR SKIP-BASED METHOD

Audio Sync Guidance (ASG) is inspired by Hyung et al. (2025) but differs in both purpose and design.

Hyung et al. (2025) improves visual fidelity by constructing a weak model that skips visually sensitive layers and using it to guide the full model. In T2I/T2V settings, however, semantic and visual features are heavily entangled, making such selective skipping difficult, model-dependent, and prone to unintended degradations.

ASG instead targets synchronization. We skip only the audio-injection layers, creating an off-sync model that preserves appearance but ignores audio cues. The difference between this off-sync and the full model isolates synchronization as the guidance signal (see Appendix B.2).

This design is suited to A2V architectures, where audio and visual pathways are explicitly separated. Skipping only the audio pathway perturbs synchronization without affecting visual fidelity, enabling precise and stable guidance for improved audio–motion alignment.

## C EVALUATION METRICS FOR SYNCHRONIZATION

### C.1 IMPLEMENTATION DETAILS OF CYCLESYNC

We use V-AURA (Viertola et al., 2025) as the pretrained video-to-audio model $f_{\text{v2a}}$, selected for its demonstrated effectiveness in generating general-class, temporally aligned audio from videos. For peak extraction, we use `librosa.onset.onset_detect`, and $\delta$ is fixed at 5 milliseconds.

Alg. 1 summarizes the full CycleSync evaluation procedure, including peak extraction and one-to-one matching.

### C.2 CONTROLLED METRIC COMPARISON

To evaluate the effectiveness of CycleSync, we compare it with three existing synchronization metrics: AV-Align (Yariv et al., 2023), AlignSync, and RelSync (Zhang et al., 2024). We apply six levels of synchronization shift to video clips from the AVSync15 (Zhang et al., 2024) and TheGreatestHits (Owens et al., 2016) datasets.

---

**Algorithm 1** CycleSync: audio reconstruction + onset-peak IoU with one-to-one matching

---

**Require:** GT audio waveform $\boldsymbol{a}$, generated video $\hat{\boldsymbol{v}}$, duration $T$ (sec), onset parameter $\Delta$, tolerance $\delta$, sampling rate $sr$
**Ensure:** CycleSync score $s \in [0, 1]$

1:  $\hat{\boldsymbol{a}} \leftarrow f_{\text{v2a}}(\hat{\boldsymbol{v}})$                                                 ▷ pretrained v2a model
2:  **function** DETECTPEAKS$(x, T, sr, \Delta)$
3:     $x \leftarrow$ RESAMPLELOAD$(x, sr)$
4:     $x \leftarrow x[0 : \lfloor sr \cdot T \rfloor]$
5:     **if** $\max |x| < 0.01$ **then**
6:         **return** $\emptyset$
7:     **end if**
8:     $e \leftarrow$ ONSETSTRENGTH$(x, sr)$
9:     $F \leftarrow$ ONSETDETECT$(e, sr, \Delta)$                 ▷ `librosa.onset.onset_detect`
10:    $P \leftarrow$ FRAMESTOTIME$(F, sr)$
11:    **return** $P$
12: **end function**
13: $\mathbb{A} \leftarrow$ DETECTPEAKS$(\boldsymbol{a}, T, sr, \Delta)$
14: $\hat{\mathbb{A}} \leftarrow$ DETECTPEAKS$(\hat{\boldsymbol{a}}, T, sr, \Delta)$
15: $I \leftarrow 0$
16: used$[1 : |\hat{\mathbb{A}}|] \leftarrow$ **false**
17: **for** $a \in \mathbb{A}$ **do**                             ▷ greedy one-to-one matching within $\delta$
18:     **for** $j \leftarrow 1$ **to** $|\hat{\mathbb{A}}|$ **do**
19:         **if** $\neg$ used$[j]$ **and** $|\hat{\mathbb{A}}[j] - a| < \delta$ **then**
20:             $I \leftarrow I + 1$
21:             used$[j] \leftarrow$ **true**
22:             **break**
23:         **end if**
24:     **end for**
25: **end for**
26: $s \leftarrow \dfrac{I}{|\mathbb{A}| + |\hat{\mathbb{A}}| - I}$
27: **return** $s$

---

**implementation detail.** We extract three 2-second clips per video with linear intervals. To ensure valid comparison under delay shifts, clips are sampled starting 0.5 seconds into the video, allowing up to 0.5 seconds of temporal shift. Videos shorter than 2.5 seconds are excluded. It results in 438 clips from 150 videos in AVSync15, and 732 clips from 244 videos in TheGreatestHits.

AlignSync and RelSync are evaluated on videos downsampled to 6 fps. AV-Align is measured at 6 fps unless otherwise noted as 24 fps. CycleSync (Ours) is evaluated at 24 fps videos.

**Synchronization configurations.** A sample type with *"Perfect Sync"* represents that Ground-truth audio pairs with its original video. The other sample types with *"Delay 0.1s–0.5s"* represent that the video is temporally shifted by the indicated delay relative to its audio.

### C.2.1 RESULTS AND ANALYSIS

Figure 7 shows how each metric responds to increasing audio-video misalignment. We observe that existing metrics often struggle to clearly separate perfectly synchronized samples from delayed ones, whereas CycleSync scores drop sharply with the misalignments. For absolute metric values, please refer to Table 5 and Table 6.

**AV-Align.** The performance of AV-Align varies significantly depending on the frame rate. At 24 fps, we would expect the highest score for perfectly synchronized samples, but in both AVSync15 and TheGreatestHits, delayed samples receive higher scores than the ground-truth alignment. At 6 fps, AV-Align becomes more stable, but the separation between perfect and delayed cases remains

limited. This suggests that the metric may not reliably reflect fine-grained temporal misalignment at higher frame rates.

Moreover, as shown in Appendix C.4, there are cases where models without audio conditioning obtain higher AV-Align scores than models explicitly guided by audio. This is because AV-Align assumes a strict one-to-one correspondence between peaks in the audio and motion signals, an assumption that often does not hold in natural scenarios, where motion may precede or follow audio cues.

**AlignSync and RelSync.**   AlignSync and RelSync generally show decreasing scores as the degree of delay increases, indicating sensitivity to misalignment. However, they do not show clear differences between perfectly synchronized samples and delayed ones, especially on TheGreatestHits dataset. In addition, both metrics are designed for evaluation at 6 fps, which makes it difficult to assess the performance of models operating at higher frame rates, such as 24 fps.

We also observe cases where models without audio conditioning receive higher scores than those guided by audio (see Appendix C.4). One possible explanation is that these metrics are more effective when evaluating sequences that are simple temporal shifts of the same ground-truth content, as assumed during training. In contrast, when the evaluated sequence differs from the original ground-truth content, the metrics may no longer provide reliable scores.

**CycleSync.**   CycleSync clearly distinguishes perfectly synchronized samples from those with temporal misalignment. Once the delay exceeds a certain threshold, the differences between misaligned cases become less pronounced. In other words, the score is not strictly monotonically decreasing.

This is due to CycleSync's use of a fixed 0.05s tolerance window to determine alignment between onset peaks in the original and reconstructed audio. While this allows for robust separation between synchronized and unsynchronized cases, it does not explicitly quantify how far misaligned peaks fall beyond the threshold.

This behavior arises because the tolerance hyperparameter is set to 0.05s, and CycleSync determines alignment between onset peaks in the original and reconstructed audio within this fixed tolerance window. While this design provides robust separation between synchronized and unsynchronized cases, it does not explicitly quantify how far misaligned peaks fall beyond the threshold. It could be addressed by incorporating multi-scale tolerance or continuous scoring mechanisms to capture varying degrees of misalignment. We leave this as future work.

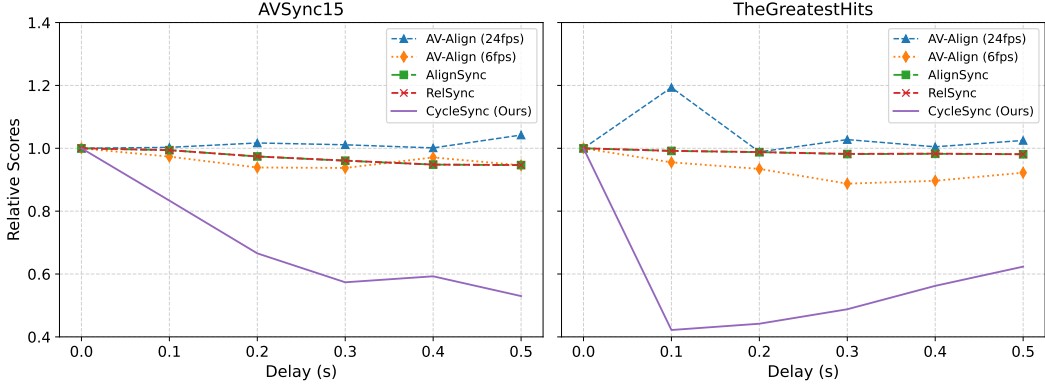

Figure 7: Comparison of relative synchronization scores under increasing audio-video delays on AVSync15 and TheGreatestHits datasets. The vertical axis denotes each metric's score normalized by its value under perfect synchronization (0.0s).

### C.3   HUMAN ALIGNMENT VALIDATION FOR CYCLESYNC

Human evaluation is essential for establishing a reliable synchronization metric. To assess how well CycleSync aligns with human perception, we conduct a user study with 9 participants, who rate the

Table 5: Comparison of synchronization metric scores on the AVSync15 dataset. Parentheses show percentage change from perfect synchronization (positive = increase, negative = decrease).

| Sample Type | AV-Align (24fps) ↑ | AV-Align ↑ | AlignSync ↑ | RelSync ↑ | CycleSync ↑ |
|---|---|---|---|---|---|
| **Perfect Sync** | 24.22 (0.0%) | **20.30** (-0.0%) | **25.04** (-0.0%) | **50.00** (0.0%) | **20.97** (0.0%) |
| Delay 0.1s | 24.29 (+0.3%) | 19.76 (-2.7%) | 24.89 (-0.6%) | 49.70 (-0.6%) | 17.48 (-16.6%) |
| Delay 0.2s | 24.63 (+1.7%) | 19.06 (-6.1%) | 24.39 (-2.6%) | 48.70 (-2.6%) | 13.96 (-33.4%) |
| Delay 0.3s | 24.49 (+1.1%) | 19.03 (-6.3%) | 24.05 (-4.0%) | 48.04 (-3.9%) | 12.03 (-42.6%) |
| Delay 0.4s | 24.25 (+0.1%) | 19.71 (-2.9%) | 23.74 (-5.2%) | 47.41 (-5.2%) | 12.43 (-40.7%) |
| Delay 0.5s | **25.24** (+4.2%) | 19.20 (-5.4%) | 23.70 (-5.4%) | 47.33 (-5.3%) | 11.11 (-47.0%) |

Table 6: Comparison of synchronization metric scores on TheGreatestHits dataset. Parentheses show percentage change from perfect synchronization (positive = increase, negative = decrease).

| Sample Type | AV-Align (24fps) ↑ | AV-Align ↑ | AlignSync ↑ | RelSync ↑ | CycleSync ↑ |
|---|---|---|---|---|---|
| **Perfect Sync** | 14.84 (0.0%) | **27.27** (0.0%) | **25.07** (0.0%) | **50.00** (0.0%) | **16.52** (0.0%) |
| Delay 0.1s | **17.71** (+19.3%) | 26.05 (-4.5%) | 24.86 (-0.8%) | 49.59 (-0.8%) | 6.97 (-57.9%) |
| Delay 0.2s | 14.67 (-1.2%) | 25.48 (-6.6%) | 24.76 (-1.2%) | 49.40 (-1.2%) | 7.30 (-55.8%) |
| Delay 0.3s | 15.25 (+2.8%) | 24.20 (-11.3%) | 24.61 (-1.8%) | 49.11 (-1.8%) | 8.06 (-51.2%) |
| Delay 0.4s | 14.91 (+0.5%) | 24.45 (-10.3%) | 24.63 (-1.8%) | 49.15 (-1.7%) | 9.29 (-43.77%) |
| Delay 0.5s | 15.21 (+2.5%) | 25.15 (-7.8%) | 24.59 (-1.9%) | 49.06 (-1.9%) | 10.30 (-37.65%) |

sync quality of 20 videos, sampled from Pyramid Flow and Syncphony, on a 1–5 scale. We then compute Pearson correlations between the human ratings and the metric scores. We compare CycleSync against existing synchronization metrics: AV-Align (Yariv et al., 2023), AlignSync, RelSync (Zhang et al., 2024), and DeSync (Cheng et al., 2025).

**Correlation with human ratings.** As shown in Table 7, CycleSync shows the strongest positive correlation with human ratings ($r = 0.486$), whereas prior metrics exhibit weak or even negative correlations. DeSync also shows a positive trend ($r = 0.206$), but remains notably weaker than CycleSync.

**Model ranking agreement.** We further compared model-level rankings derived from each metric against human ratings (Table 8). CycleSync accurately captures the human-preferred ordering between the two models. DeSync also reflects the correct direction of preference, though with lower discriminative strength.

These results provide strong empirical evidence that CycleSync is both quantitatively sensitive to temporal misalignment and best aligned with human perception, making it a more reliable synchronization metric than existing metrics.

## C.4 RESULTS OF BASELINES AND SYNCPHONY WITH EXISTING METRICS

We additionally report the performance of baseline models and ours using existing synchronization metrics on the AVSync15 and TheGreatestHits datasets in Table 9 and Table 10, respectively.

On AVSync15, the fine-tuned Pyramid Flow model, which generates audio-independent but plausible motion, achieves the highest AV-Align score. A similar pattern is observed in TheGreatestHits, where the same model also obtains higher AlignSync and RelSync scores than other audio-conditioned models.

These results reveal a limitation of existing metrics, which tend to favor models that produce highly dynamic motion with plausible timing, even if that motion is not aligned with the audio signal.

In contrast, CycleSync consistently assigns the lowest scores to the same model across both datasets. This is because CycleSync penalizes mismatches in temporal structure between the original audio and the reconstructed audio from the generated video. Rather than comparing audio and motion

Table 7: **Correlation with human ratings.** CycleSync achieves the highest positive correlation with human judgments, while other metrics show weak or negative trends.

| Metric | Correlation | 95% CI Lower | 95% CI Upper |
|---|---|---|---|
| CycleSync | **0.486** | 0.053 | 0.919 |
| AV-Align | 0.043 | -0.451 | 0.538 |
| RelSync | -0.623 | -1.011 | -0.236 |
| AlignSync | -0.625 | -1.011 | -0.238 |
| DeSync | 0.206 | -0.279 | 0.690 |

Table 8: **Model ranking agreement with human ratings.** CycleSync correctly reflects human preference, ranking Syncphony above Pyramid Flow.

| Model | Human Score ↑ | CycleSync ↑ | AV-Align ↑ | RelSync ↑ | AlignSync ↑ | DeSync ↓ |
|---|---|---|---|---|---|---|
| Pyramid Flow (I2V) | 2.68 | 8.15 | **24.96** | **55.36** | **27.75** | 0.88 |
| Syncphony (Ours) | **4.30** | **22.04** | 21.88 | 50.44 | 25.19 | **0.86** |

peaks directly, CycleSync compares the temporal structure of the original and reconstructed audio signals, enabling more precise assessment of synchronization quality.

### C.5 LIMITATION OF CYCLESYNC.

As a reconstruction-based metric, CycleSync relies on the quality and behavior of the underlying video-to-audio (V2A) model. The reconstructed audio may sometimes reflect dataset-level biases rather than the visual content of the input video itself.

For example, in frog videos, although only a single frog may be visible, many clips in the dataset include ambient sounds from nearby frogs. As a result, the reconstructed audio sometimes contains multiple frog sounds, regardless of the actual motion in the video. Similarly, bowling videos in the dataset often include background music, which can occasionally appear in the reconstructed audio even if it is not visually implied. Such cases may affect CycleSync scores in specific contexts. This issue could potentially be addressed by improving the V2A model or applying post-processing, which we leave for future work.

## D ARCHITECTURAL DETAILS

### D.1 TRAINING LAYER SELECTION

#### D.1.1 VIDEO GENERATION BACKBONE

We adopt Pyramid Flow (Jin et al., 2024a) as the video generation backbone due to its efficiency and scalability in generating long, high-resolution videos. Pyramid Flow is an autoregressive diffusion transformer trained with a flow-matching objective, which sequentially synthesizes consecutive video chunks by denoising each chunk for given a previous chunk and a text prompt.

To capture temporal and spatial consistency, it employs 3D Rotary Positional Encoding (RoPE) (Su et al., 2024) within its self-attention layers, enabling the model to encode relative positions across time, height, and width. In addition, the model dynamically adjusts resolution throughout the denoising process, using low-resolution frames at early (noisier) timesteps and high-resolution frames at later (cleaner) stages, thereby reducing computational cost while maintaining visual detail.

This design enables resource-efficient training and generation, supporting high-resolution and long-duration video synthesis even under constrained computational resources.

Table 9: Quantitative results on the AVSync15 dataset.

| Input | Model | AV-Align ↑ | AlignSync ↑ | RelSync ↑ | CycleSync ↑ |
|---|---|---|---|---|---|
| T+A | TempoTokens (Yariv et al., 2023) | 15.51 | 22.38 | 46.91 | 13.10 |
| I+T | Pyramid Flow (Jin et al., 2024a) | 18.85 | 23.65 | 47.56 | 14.25 |
| | Pyramid Flow (fine-tuned) | **20.69** | 23.97 | 47.76 | 12.34 |
| I+T+A | AVSyncD (Zhang et al., 2024) | 19.31 | **24.61** | 48.99 | 16.38 |
| | Ours w/o ASG ($w_a = 0$) | 20.01 | 24.24 | 48.28 | 15.31 |
| | Ours w/ ASG ($w_a = 2$) | 19.89 | 24.45 | 48.74 | **16.48** |
| | Ours w/ ASG ($w_a = 4$) | 20.00 | 24.58 | **49.04** | 16.26 |
| | *Groundtruth* | 20.84 | 25.10 | 50.00 | 22.15 |

Table 10: Quantitative results on TheGreatestHits dataset.

| Input | Model | AV-Align ↑ | AlignSync ↑ | RelSync ↑ | CycleSync ↑ |
|---|---|---|---|---|---|
| I+T | Pyramid Flow (Jin et al., 2024a) | 25.24 | 25.12 | 50.46 | 7.41 |
| | Pyramid Flow (fine-tuned) | 26.76 | 26.67 | 53.35 | 9.23 |
| I+T+A | AVSyncD (Zhang et al., 2024) | 23.29 | 26.55 | 53.07 | 9.89 |
| | Ours w/o ASG ($w_a = 0$) | **27.11** | 26.08 | 52.21 | 11.70 |
| | Ours w/ ASG ($w_a = 2$) | 26.92 | 26.10 | 52.27 | **16.18** |
| | Ours w/ ASG ($w_a = 4$) | 26.81 | **27.04** | **54.14** | 17.71 |
| | *Groundtruth* | 26.00 | 25.07 | 50.00 | 15.99 |

### D.1.2 TRAINING LAYER SELECTION IN VIDEO BACKBONE

Pyramid Flow consists of 24 transformer blocks. To identify which layers to fine-tune, we individually skip each of the 24 transformer blocks during inference and observe the effects on image-to-video (I2V) generation (see Figures 10 and 11).

Skipping early blocks (0–7) significantly degrades appearance, often causing artifacts in the background and object structure. In contrast, skipping later blocks (8–23) mostly preserves the appearance of the input image (first frame) in the generated video, primarily affecting the motion. This suggests that early blocks are critical for preserving the input's appearance, whereas later blocks are responsible for refining motion. This separation aligns with the architecture: early blocks use separate attention weights for text and video, while later blocks share them. Based on this functional and structural separation, we fine-tune only the last 16 blocks (8–23) with minimal impact on the pretrained model's visual fidelity.

### D.2 AUDIO RoPE

### D.2.1 IMPLEMENTATION DETAILS

To encode the temporal structure of audio features explicitly, we apply **Rotary Positional Encoding (RoPE)** to inject relative temporal information directly into the cross-attention mechanism, as illustrated in Figure 3(a).

We first obtain video latents $\{z^{(l)}\}_{l=0}^{L_{\text{video}}}$ from a VAE encoder, where each $z^{(l)}$ represents a compressed spatiotemporal feature at the $l$-th position in the video sequence. Simultaneously, we extract audio features $\{a^{(i)}\}_{i=0}^{L_{\text{audio}}}$ from a DenseAV encoder, capturing the temporal and semantic structure of the audio input.

To align these modalities, we divide the audio sequence into local segments corresponding to each video latent. For each target video latent $z^{(l)}$, we define the corresponding audio segment $\mathbb{A}^{(l)}$ as:

$$\mathbb{A}^{(l)} = \{a^{(i)} \mid i \in [\alpha(l - \Delta), \alpha(l + \Delta)]\}, \tag{11}$$

Table 11: Ablation of Audio RoPE.

| Model Variant | CycleSync ↑ |
|---|---|
| w/o Audio RoPE | $14.41_{\pm 1.40}$ |
| w/ Audio RoPE | $15.31_{\pm 1.49}$ |

where $\alpha$ is a scaling factor mapping video indices to audio indices (accounting for the different sequence lengths), and $\Delta$ determines the width of the temporal window(we set $\Delta=1$).

Then, we apply Audio RoPE to the audio segments. The procedure is as follows:

**Step 1. Assign positional indices.**

- Each video latent $z^{(l)}$ is assigned 3D coordinates $(l, h, w)$ representing its temporal and spatial location within the video sequence.
- For each audio segment $\mathbb{A}^{(l)}$, the constituent audio features are assigned linearly interpolated temporal indices within the range $[l - (\Delta + 0.5), l + (\Delta + 0.5)]$, such that the center of the segment aligns exactly with $l$.

**Step 2. Project into query and key spaces.**

$$q^{(l)} = W_Q z^{(l)}, \quad K^{(l)} = \{W_K a^{(i)} \mid a^{(i)} \in \mathbb{A}^{(l)}\}, \tag{12}$$

where $W_Q$ and $W_K$ are learnable linear projection matrices.

**Step 3. Apply RoPE rotations.**

$$q_{\text{rope}}^{(l)} = \text{RoPE}(q^{(l)}, (l, h^{(l)}, w^{(l)})), \quad K_{\text{rope}}^{(l)} = \{\text{RoPE}(W_K a^{(i)}, (\tau^{(i)}, 0, 0)) \mid a^{(i)} \in \mathbb{A}^{(l)}\}, \tag{13}$$

where $\tau^{(i)}$ denotes the interpolated temporal index assigned to each $a^{(i)}$.

**Step 4. Compute cross-attention between video latent $z^{(l)}$ and audio segment $\mathbb{A}^{(l)}$:**

$$\text{Attention}(z^{(l)}, \mathbb{A}^{(l)}) = \text{Softmax}\left(\frac{q_{\text{rope}}^{(l)}(K_{\text{rope}}^{(l)})^\top}{\sqrt{d}}\right) z^{(l)}, \tag{14}$$

where $z^{(l)} = \{W_V a^{(i)} \mid a^{(i)} \in \mathbb{A}^{(l)}\}$ is the set of value projections of the audio features, and $d$ is the dimension of the projected space.

By explicitly injecting temporally aligned positional cues into both video and audio features, our model captures the sequential structure of audio signals more effectively, leading to improved synchronization between generated video motion and corresponding audio events.

### D.3 ABLATION STUDY

We conduct ablation experiments to examine the effect of Audio RoPE. The results in Table 11 indicate that using Audio RoPE leads to higher synchronization quality compared to the model without it. Without applying RoPE to audio features, the model frequently exhibits misalignments, with motions often preceding or lagging behind the corresponding audio cues. In contrast, applying RoPE to the audio features results in tighter temporal alignment between motion and sound events, enabling the model to better capture the sequential structure of the audio input. Additional ablation examples are included in the supplementary materials (Appendix G).

## E USER STUDY

To assess the perceptual quality of our generated videos, we conducted a user study comparing our method with the state-of-the-art Audio-to-Video model AVSyncD (Zhang et al., 2024). We select

AVSyncD as the sole baseline in the user study, as other baselines generate noticeably unsynchronized motion. The evaluation focused on three aspects: synchronization with audio, image quality, and frame consistency.

The study was conducted using all 150 test videos from the AVSync15 dataset. These were divided into five subsets of 30 videos each, with each subset assigned to two participants (10 participants total). For every video, participants were shown two versions, one generated by our model and one by AVSyncD based on the same audio input and initial image. Participants were asked to answer the three questions for each video pair:

- **Synchronization:** Which video is better synchronized with the audio in terms of motion timing?

- **Image Quality:** Which video has better image quality in terms of realism and clarity?

- **Frame Consistency:** Which video is more visually consistent across frames, without flickering or sudden jumps?

As illustrated in Figure 9, participants evaluated video pairs using a web interface showing both videos and three corresponding questions.

The results, summarized in Figure 8, show that our model was consistently preferred: 74% for synchronization, 90% for image quality, and 94% for frame consistency.

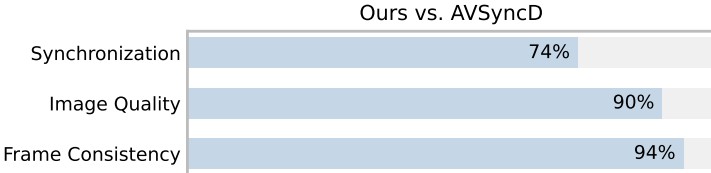

Figure 8: Human preference rates (%) for our method over ASyncD across three evaluation criteria.

These results demonstrate that our model is consistently favored by human evaluators across all three aspects. This further validates the effectiveness of our synchronization mechanisms and the visual fidelity of our methods.

## F    ADDITIONAL VIDEO SAMPLES

Additional generated videos, ablations, and comparisons are provided on our project page:

https://jibin86.github.io/syncphony_project_page.

### F.1    SAMPLES FROM SYNCPHONY

Please refer to our project page for generated video samples from Syncphony.

## G    ABLATION SAMPLES

Please refer to Section **Ablations** of the project page , which includes ablation results for Motion-aware Loss, Audio Sync Guidance, and Audio RoPE.

## H    COMPARISON SAMPLES

Please refer to Section **Comparison** of the project page for video comparisons between our model, AVSyncD Zhang et al. (2024), and Pyramid Flow (fine-tuned), a variant of our model without audio cross-attention layers.

# I IMPLEMENTATION AND EXPERIMENTAL DETAILS

## I.1 WHY IMAGE-TO-VIDEO BACKBONE?

We also applied our method to a Text-to-Video (T2V) model, AnimateDiff (Guo et al., 2023), and trained it on the AVSync15 dataset, which contains limited 1,350 training samples. We found the model generates motion aligned with audio, but it shows overfitting, with limited diversity in appearance. This is because T2V models have to generate both appearance and motion without a reference image. With a small dataset, it becomes difficult to produce diverse appearances, and even harder to learn various audio-driven motion patterns.

In contrast, Image-to-Video (I2V) models, such as Pyramid Flow (Jin et al., 2024a), are conditioned on an initial image and focus on predicting motion rather than full appearance. This simplifies the learning process and reduces the risk of overfitting. For these reasons, we adopt the I2V model as our video generation backbone.

## I.2 TRAINING AND INFERENCE SETTINGS

We train our model using 4 NVIDIA RTX 3090 GPUs (24GB each) with a total batch size of 32. Training takes 34 hours to reach 33,000 steps. For all experiments, we use 30 denoising steps. We follow Pyramid Flow in setting the classifier-free guidance (CFG) strength to 7.0 for the first latent and 4.0 for the rest. For Audio Sync Guidance, we use $w = 2$, where $w = 0$ disables the guidance.

Inference time for a 5-second video (with pre-encoded audio and text features) is as follows:

- Audio Guidance: 2 min 53 sec
- w/o Audio Guidance: 2 min 01 sec
- w/o Audio Layers: 1 min 43 sec

At least 16 GB of GPU memory is required to generate 5-second videos.

## I.3 TRAINING AND EVALUATION DATASETS

We use two datasets for training and evaluation:

- **AVSync15 (Zhang et al., 2024):** 1,350 videos for training and 150 for testing. For evaluation, we linearly extract 3 clips per video, resulting in 450 evaluation clips.
- **TheGreatestHits (Owens et al., 2016):** 733 videos for training and 244 for testing, resulting in 732 evaluation clips.

During training, we randomly sample clips from different temporal regions of each video to improve generalization to various audio-motion alignments.

# J APPLICABILITY OF SYNCPHONY TECHNIQUES TO OTHER MODALITIES

While Syncphony focuses on audio-to-video generation, we believe the proposed techniques are applicable to other modalities.

Motion-aware Loss, by amplifying learning signals in high-motion regions, encourages the model to focus on dynamic cues that reflect physically grounded movements. This can benefit tasks like audio-to-3D animation, text-to-video, and text-to-3D, where generating realistic motion is essential.

In contrast, Audio Sync Guidance are designed to improve synchronization between audio and motion. This technique is applicable to tasks such as audio-to-3D animation, provided that the model adopts an attention-based architecture with functionally well-separated layers, which enables clean injection of audio signals into the network.

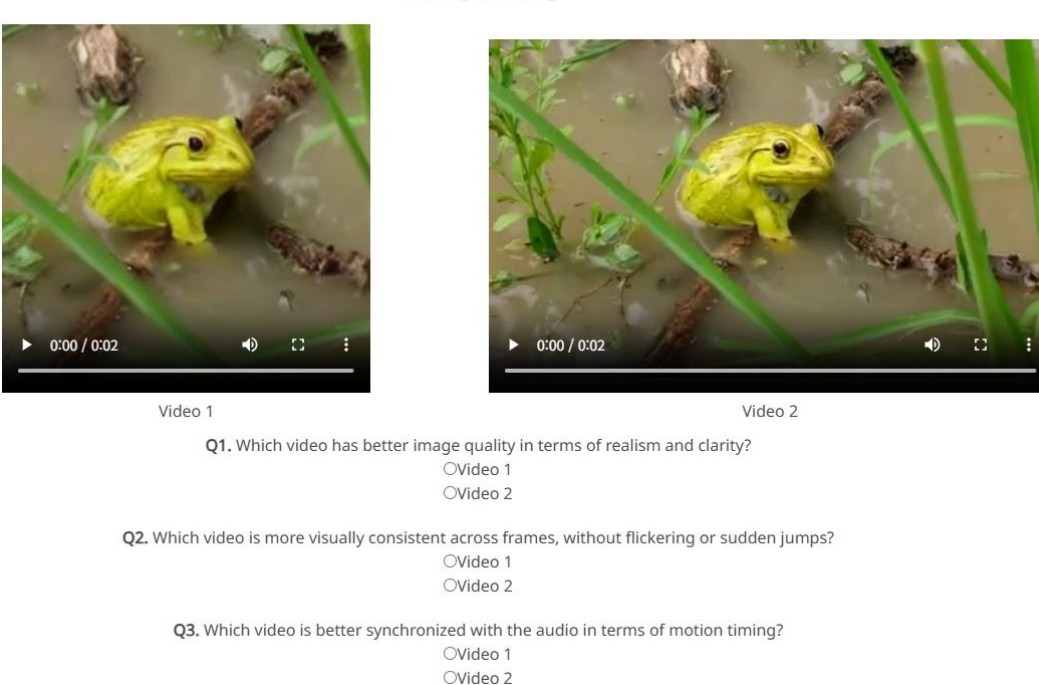

Figure 9: Screenshot of the user study interface of each video pair with questions.

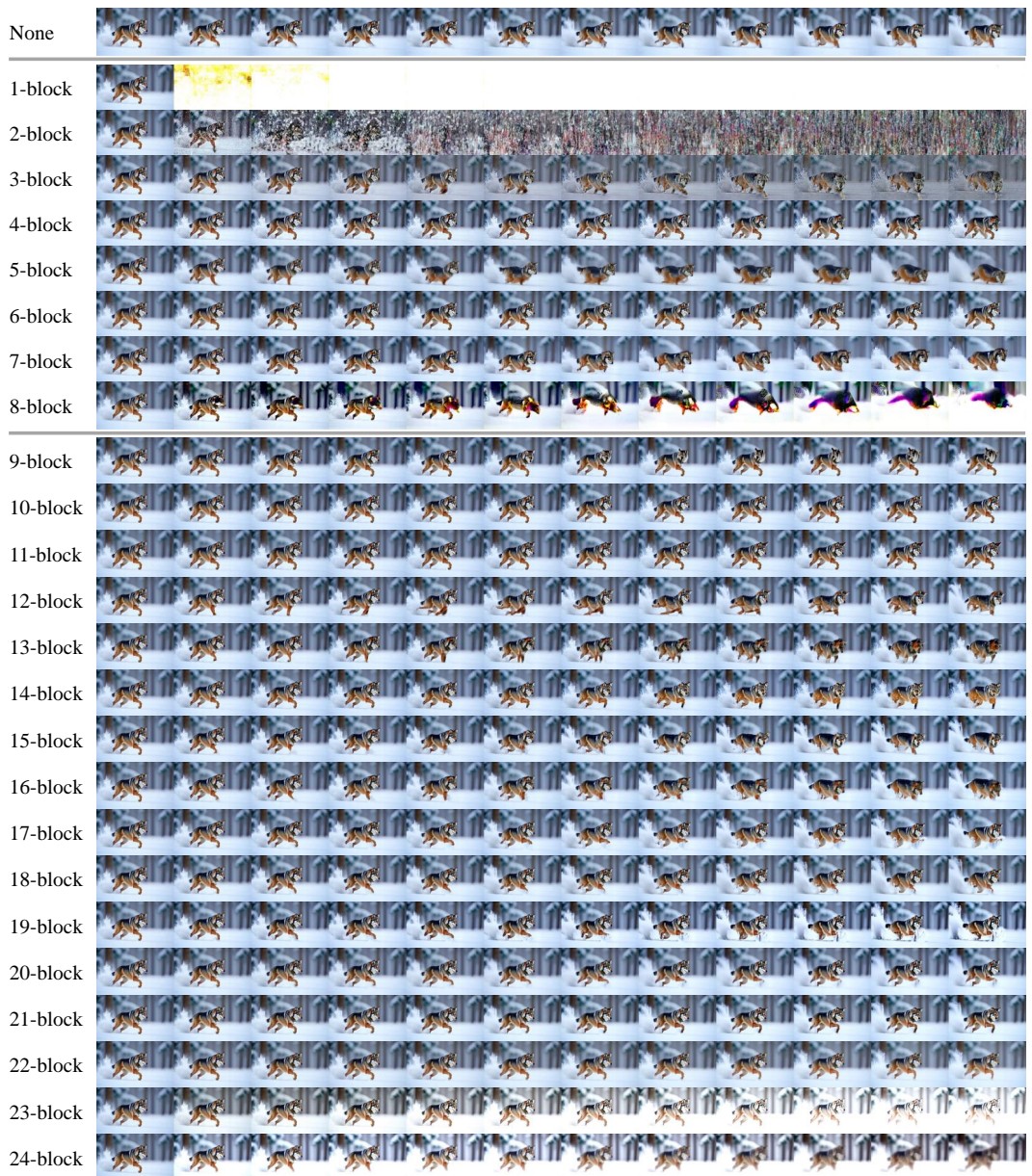

Figure 10: Frame results of skipping each transformer block individually.

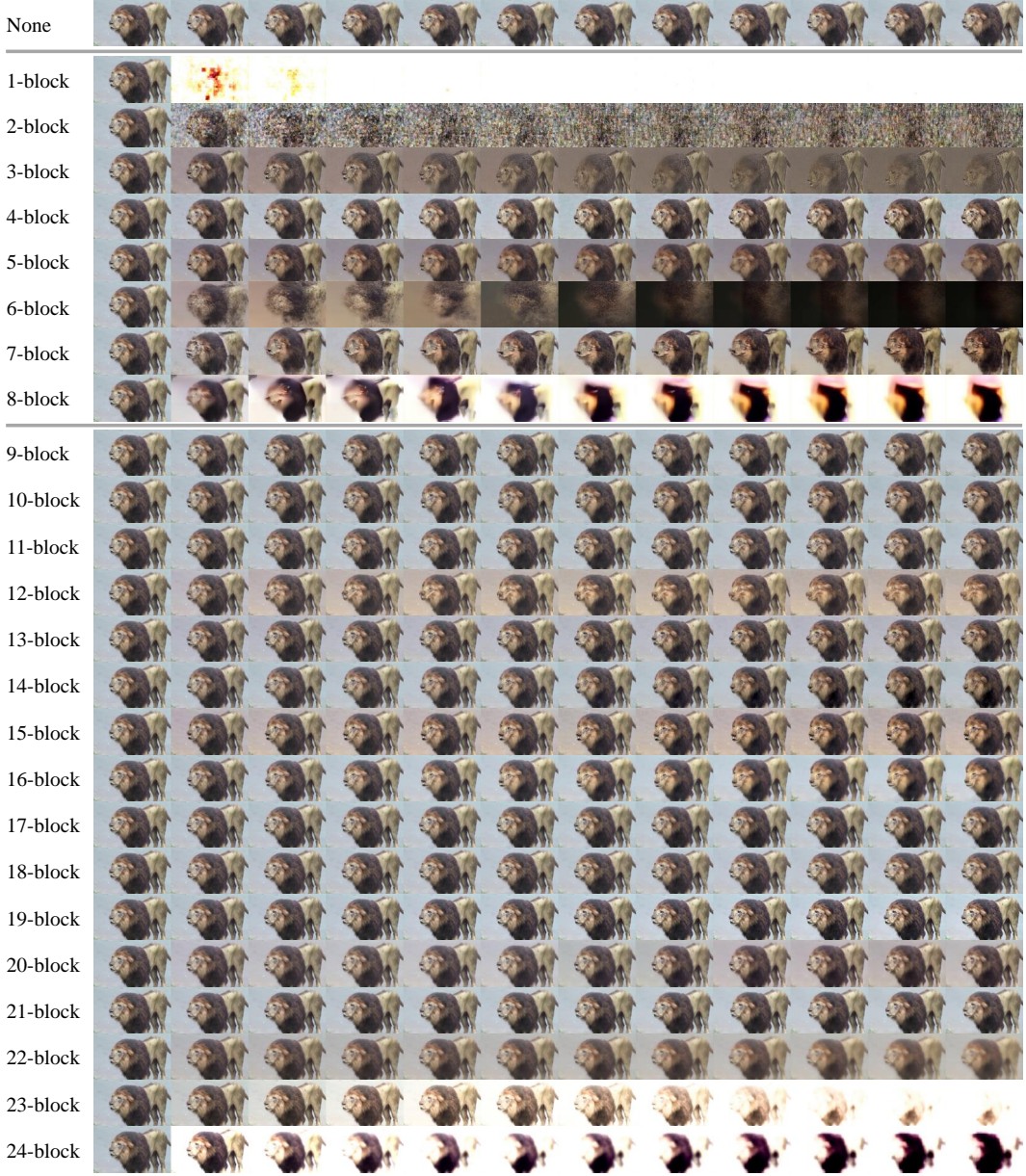

Figure 11: Frame results of skipping each transformer block individually.

