# OpenReview forum: "Syncphony: Synchronized Audio-to-Video Generation with Diffusion Transformers"
_ICLR.cc/2026/Conference — ICLR 2026 Poster_

### Official Review · Reviewer_2TxV · 2025-10-31

**Soundness:** 3
**Presentation:** 3
**Contribution:** 2
**Rating:** 4
**Confidence:** 4

**Summary:**

The paper proposes Syncphony, an audio-to-video generation model that focuses on improving synchronization between audio and motion.
To achieve this, the authors introduce a Motion-Aware Loss for capturing movement intensity and an Audio Sync Guidance (ASG) mechanism for enforcing synchronization between audio and visual dynamics.
A new evaluation metric, CycleSync, is also proposed to better align with human perceptual judgments of synchronization.
Experimental results show that Syncphony performs better than most existing baselines.

**Strengths:**

1.	The introduction of the CycleSync metric sounds reasonable and shows higher alignment with human perceptual evaluations than prior synchronization metrics.
2.	The model achieves consistently better performance than most baselines across several benchmarks.

**Weaknesses:**

1.	Generating a 5-second video takes nearly 3 minutes, which significantly limits practical usability.
2.	The main model builds heavily on Pyramid Flow, with only moderate extensions (audio conditioning and synchronization guidance). As such, the contributions feel incremental.
3.	The CycleSync metric relies on pretrained V2A models that may introduce background or irrelevant audio content, potentially biasing the evaluation.
4.	The proposed Motion-Aware Loss does not explicitly capture semantic motion as authors mentions. It merely measures pixel or latent differences between consecutive frames, which may not correspond to meaningful sound-related motion.

**Questions:**

In the demo, only 2-second video samples are provided, even if it can generate 5 seconds video. Why is that?

---

> ### Author Response · Authors · 2025-11-21
> **Response to 2TxV**
>
> We sincerely thank the reviewer for the thoughtful evaluation. We appreciate the recognition that CycleSync provides a more perceptually aligned synchronization metric, and that our model achieves consistently strong performance across multiple benchmarks. We are grateful that these key contributions were acknowledged.
>
> Below, we provide a detailed response that addresses the reviewer’s concerns.
>
> ### (W1) Inference Speed Considerations
> Fast sampling goes beyond the scope of this work and is generally handled by a separate line of research. Recent studies have accelerated video models by several-fold. Likewise, if existing acceleration techniques are applied to our model, its inference time can also be significantly improved.
>
> ### (W2) Design Intent and Backbone Flexibility
>
> We would like to clarify (1) our design intent and (2) that the proposed methods are not tied to the Pyramid Flow backbone.
>
> **(1) Design intent**
>
> As we kindly clarify in lines L017, L053, and L302, our goal is not to propose a new video-generation backbone, but to introduce mechanisms for audio-visual synchronization on top of existing pretrained video models.
>
> Importantly, leveraging a pretrained backbone is not a lightweight modification. Training a video diffusion transformer from scratch is extremely resource-intensive, and naively fine-tuning the entire backbone degrades the pretrained model’s spatial and temporal generation quality. Therefore, the challenge is to introduce audio-motion synchronization without compromising the strong visual fidelity and temporal coherence already learned by the backbone. Syncphony is designed to address this challenge.
>
> **(2) The proposed methods are not tied to the Pyramid Flow backbone**
>
> In addition, we further emphasize that the proposed Motion-Aware Loss and Audio Sync Guidance, designed for audio-visual synchronization on pretrained backbones, are not tied to Pyramid Flow. We believe there is no architectural constraint that would prevent these mechanisms from being applied to other pretrained video models.
>
> Motion-Aware Loss computes motion magnitude from ground-truth consecutive frames and uses it to reweight the training objective, enabling the model to focus more on motion-relevant regions. This formulation is not tied to Pyramid Flow’s autoregressive structure; it could be applied equally to any diffusion/flow based video models, by weighting each frame’s prediction error using its corresponding ground-truth motion magnitude.
>
> Similarly, Audio Sync Guidance is designed for video models where visual generation and audio-conditioning pathways are functionally separated. In such architectures, ASG offers a clean and stable way to modulate the strength of audio cross-attention, and thus may extend beyond Pyramid Flow.
> In addition, as described in Appendix B.1, conventional classifier-free guidance (CFG) is trained by randomly dropping conditioning inputs as zero. However, we observe that directly applying this random-drop strategy to audio conditioning leads to degraded performance.
> Audio carries a meaningful semantic interpretation even when its value is dropped; silence is itself an informative condition. To address this issue, instead of dropping audio conditions, we propose an Audio Sync Guidance.
>
> In summary, our contributions do not depend on Pyramid Flow, and the proposed mechanisms may be applicable to other pretrained video models as well.
>
>
> ### (W3) About CycleSync
>
> As reviewer DdqZ mentioned, CycleSync offers a meaningful step forward over prior metrics (AV-Align, AlignSync, RelSync) by better correlating with human perception.
> CycleSync introduces a novel evaluation perspective by measuring whether the generated video contains sufficient timing and motion cues to reproduce the original audio structure (line 323), a capability not provided by existing metrics. As shown in line 431, CycleSync achieves the highest correlation with human judgments among all metrics we compare against.
>
> As discussed in Appendix C.4 (line 858), existing metrics often favor models that produce highly dynamic motion with plausible timing, even when the motion is not actually aligned with the audio signal. In contrast, CycleSync explicitly penalizes such audio-motion mismatches, allowing it to detect misalignment cases that existing metrics couldn't.
>
> We acknowledge, as noted in line 485 and Appendix C.5, that V2A models may introduce biased audios. However, our experiments show that CycleSync is able to capture misalignment patterns that other metrics fail to recognize. Moreover, as V2A models continue to improve, we expect CycleSync’s reliability and discriminative power to further increase.

---

> ### Author Response · Authors · 2025-11-21
>
> ### (W4) How Motion-Aware Loss Supports Sound-Related Motion Alignment
> Thank you for raising this concern. We first kindly clarify that even if the latent differences between consecutive frames do not explicitly encode semantic sound-related motion, this does not prevent the model from learning meaningful sound-related motion.
>
> As described in line 240, Motion-Aware Loss computes the magnitude of ground-truth motion between consecutive frames and uses it to reweight the loss. This mechanism strengthens the learning signal specifically in motion regions, encouraging the model to focus on dynamic areas and generate them more accurately.
>
> While doing so, the model learns the cause of each motion.
>
> If the motion is caused by audio, the model learns to associate it with the audio; if the motion is unrelated to audio (e.g., background motion, camera movement), the model learns to treat it as sound-unrelated motion. In this way, the model generates synchronized sound-related motion appropriately while still generating sound-unrelated motion.
>
> The training dataset we used includes various camera motions such as zoom-in/zoom-out. Our generated results preserve these sound-unrelated motions while synchronizing only the sound-related parts. For example, in the “dog barking” sample in the Comparison section of our demo, the dog barks in sync with the audio while the camera continues to zoom, illustrating the model’s ability to distinguish sound-driven from non-sound-driven motion.
>
> In summary, although the latent differences themselves do not directly represent semantic motion, Motion-Aware Loss effectively guides the model to learn meaningful audio-motion relationships. We will clarify these points in the revised version.
>
> ### (Q1) Demo Videos
> We will add the 5-second samples to the demo page and share the link shortly.

---

> ### Comment · Reviewer_2TxV · 2025-11-25
>
> I appreciate the authors’ detailed responses. However, several of my key concerns remain insufficiently addressed.
>
> **(1) Reliance on a single backbone (Pyramid Flow)**
>
> The authors state that their method is not tied to PyramidFlow and can be applied to other video backbones. However, all experiments are conducted only on PyramidFlow. Without evidence, it remains unclear whether Pyramid Flow is uniquely suited for this method or whether the proposed components are generally applicable across different architectures.
> Do the authors have experiments or analyses demonstrating that the proposed modules (e.g., ASG) remain effective on other video generation backbones?
> Given that Pyramid Flow is already strong after fine-tuning (Table 1 and Table 2), adding audio conditioning on top of it may appear incremental without such validation.
>
> **(2) Efficiency vs. marginal improvements**
>
> The authors argue that inference speed is orthogonal to the contribution, but this does not fully address my concern. According to the paper, generating a 5-second clip takes ~3 minutes, nearly double the time of the baseline and ~1 minute slower even without ASG.
> However, in Table 1, the performance improvements from adding audio are relatively marginal considering this. What's the author's thoughts?
> Also, how's the inference speed for AVSyncD and TempoTokens?
>
> **(3) Conventional classifier-free guidance (CFG)**
>
> The authors claim CFG “does not work” for audio because silence itself is an informative condition. However, this argument is not fully convincing:
>
> - In CFG, the null condition is not required to be semantically natural. It is simply an artificial unconditioned distribution used for guidance.
> - Prior work, such as AudioLDM [1], successfully employs CFG by dropping 10% of audio inputs to train unconditional diffusion models. (Section 3.4. Classifier-free guidance in AudioLDM paper).
> - The paper does not provide quantitative results comparing conventional CFG vs. ASG. (If I missed such an experiment, please point me to it.)
>
> [1] AudioLDM: Text-to-Audio Generation with Latent Diffusion Models (ICML 2023)
>
> **(4) CycleSync**
>
> The human study supporting CycleSync seems too limited to draw strong conclusion (only nine participants and 20 videos), which is a very small evaluation scale for a perceptual synchronization metric. In addition, as shown in Table 6 (Appendix), CycleSync appears less robust than other metrics under temporal delay on TheGreatestHits dataset, which specifically requires accurate audio–motion temporal alignment. The metric shows large fluctuations and inconsistent separation across delay levels.

---

> > ### Author Response · Authors · 2025-12-03
> >
> > Thank you for the thoughtful follow-up questions. Since real-time discussion is not possible due to OpenReview constraints, we provide detailed clarifications addressing all remaining concerns.
> >
> > ## **(1) Reliance on a single backbone (Pyramid Flow)**
> >
> > We would first like to clarify a potential misunderstanding regarding Table 1.
> >
> > The “Pyramid Flow (fine-tuned)” results refer to a model fine-tuned on AVSync15 video without audio.
> >
> > We reported this for two reasons:
> >
> > **(a) Video quality comparison.**
> >
> > The fine-tuned Pyramid Flow achieves FID/FVD scores comparable to ours, demonstrating that our method does not degrade visual quality, even after adding audio conditioning and ASG.
> >
> > **(b) Metric instability of prior audio-video metrics.**
> >
> > Despite receiving *no audio conditioning*, the fine-tuned Pyramid Flow baseline surprisingly achieves higher scores on existing synchronization metrics than some of A2V models. This behavior is inconsistent with the fact that the model is not capable of audio–motion synchronization, whereas CycleSync assigns this baseline a correctly low score. This supports our argument that existing metrics are fundamentally unstable and motivates the need for CycleSync (Appendix C.4).
> >
> > Again, we want to clarify that “Pyramid Flow (fine-tuned)” doesn’t have any audio signal itself and is not *strong*.
> >
> > **New experiments on other backbones**
> >
> > Nonetheless, we fully acknowledge the reviewer’s request and conducted additional experiments on two other video generation backbones:
> >
> > 1. **WAN-2.2 S2V**
> >
> >     We compared conventional zero-audio CFG vs. our ASG.
> >
> >     ASG consistently produced videos with stronger lip motion and clearer audio-dependent temporal cues.
> >
> > 2. **ASVA**
> >
> >     Again, ASG improved audio responsiveness over zero-audio CFG.
> >
> >
> > We provide qualitative comparisons on our project page: [B. Application of ASG to Other A2V Model](https://anonymous-submission12338nci.github.io/random-random-supplementary-web/).
> >
> > These results demonstrate that **ASG is architecture-agnostic and consistently improves audio alignment**, even on different models.
> >
> > ## **(2) Efficiency vs. marginal improvements**
> >
> > We clarify the numbers in Appendix I.2:
> >
> > For 30 denoising steps:
> >
> > - Baseline Pyramid Flow: 103s
> > - Ours w/o ASG: 121s (≈ +0.6s per step)
> > - Ours w/ ASG: 173s
> >
> > The additional cost from audio layers is modest (~0.6s per step), and the higher cost of ASG arises because ASG requires three forward passes, similar to PAG [1]/SAG [2] techniques widely used for quality boosting in image models.
> >
> > Acceleration methods for diffusion/flow sampling (many of which speed up models by ×2–×20) are orthogonal to this work and can be applied to ours as well.
> >
> > **Inference Speed Comparison (Ours vs. AVSyncD vs. TempoTokens)**
> >
> > Per reviewer request, we report the speeds (same hardware):
> >
> > | **Method** | #Frames | FPS | Audio Duration | Resolution | **Generation Time** |
> > | --- | --- | --- | --- | --- | --- |
> > | **ASVA** | 12 | 6 fps | 2 sec | 256×256 | **24 sec** |
> > | **TempoTokens** | 24 | 12 fps | 2 sec | 384×384 | **24 sec** |
> > | **Syncphony (Ours)** | 49 | 24 fps | 2 sec | 640×384 | **61 sec** |
> >
> > Notably, when normalized by *number of frames and resolution*, our method is *faster than TempoTokens and ASVA even if we use ASG*. Thus, our approach is computationally competitive while producing the highest resolution among all methods.
> >
> > [1] PAG: Self-Rectifying Diffusion Sampling with Perturbed-Attention Guidance
> >
> > [2] SAG: Improving Sample Quality of Diffusion Models Using Self-Attention Guidance

---

> ### Author Response · Authors · 2025-12-03
>
> ## **(3) Conventional Classifier-Free Guidance (CFG)**
>
> ### **(a) ASG vs CFG (New Experiments)**
>
> As shown in the WAN-2.2 S2V and ASVA experiments, *ASG outperforms CFG in practice*. ([B. Application of ASG to Other A2V Model](https://anonymous-submission12338nci.github.io/random-random-supplementary-web/))
>
> These results directly address the reviewer’s request: we provide CFG vs ASG comparisons, and ASG consistently yields stronger audio responsiveness.
>
> ### **(b) Why dropping audio fails specifically for Audio-to-Video**
>
> **(b-1)**
>
> We fully agree with the reviewer that CFG does not require the null condition to be semantically natural.
>
> However, this does *not* imply the reverse; that some distribution, including a semantically meaningful one, can serve as the null condition.
>
> The issue in A2V is exactly this:
>
> the dropped-audio representation used by CFG corresponds to silence, which is itself a highly meaningful and structured condition in audio-to-video generation.
>
> Thus, CFG forces the model to treat a *semantically meaningful* signal (“silence”) as if it were an *uninformative* unconditional branch, violating the core assumption of CFG and leading to degraded behavior in A2V.
>
> **(b-2)**
>
> This issue is specific to A2V and does *not* apply to T2A (text-to-audio) models like AudioLDM. In text-to-audio, dropping the audio latent is equivalent to “no audio condition,” not “silence,” and is therefore harmless.
>
> **(b-3)**
>
> We also emphasize that this was not a claim made in the main paper; it was provided during the rebuttal period only to clarify the motivation behind ASG in response to the reviewer’s question. Nonetheless, both the intuition and the new experiments support the reviewer’s concern that simple audio dropping is problematic in A2V.
>
> ## **(4) CycleSync**
>
> ### **(a) Human study scale**
>
> We expanded our human evaluation significantly:
>
> - Participants: **from 9 → 37**
> - Videos: **from 20 → 40**
>
> The expanded results strengthen the correlation between CycleSync and human preference and further highlight the weaknesses of prior metrics.
>
> We report the updated results in the revised appendix.
>
> **Model Ranking Agreement (Appendix C.3 Table 8):**
>
> | Model | Human Score ↑ | CycleSync ↑ | AV-Align ↑ | RelSync ↑ |
> | --- | --- | --- | --- | --- |
> | Pyramid Flow (I2V) | 2.032432 | 8.69697 | **24.963203** | **53.656925** |
> | Syncphony (Ours) | **4.45** | **24.694611** | 22.522367 | 49.269017 |
>
> **Correlation with Human Ratings (Appendix C.3 Table 7):**
>
> | Metric | Correlation | 95% CI Lower | 95% CI Upper |
> | --- | --- | --- | --- |
> | Human Score | 1.000 | 1.000 | 1.000 |
> | CycleSync | **0.620** | **0.362** | **0.878** |
> | AV-Align | 0.069 | -0.259 | 0.397 |
> | RelSync | -0.426 | -0.723 | -0.129 |
>
> ### **(b) Performance under temporal delay on TheGreatestHits dataset**
>
> We appreciate the reviewer’s careful reading of Table 6.
>
> We agree that in Table 6, CycleSync does not show a perfectly monotonic trend under *fine-grained* temporal-delay perturbations on TheGreatestHits dataset. Importantly, this behavior stems from the nature of the dataset itself rather than from the instability of the metric.
>
> TheGreatestHits consists of repeated percussive impacts with variable spacing.
> In such rhythmic sequences, artificially delaying the audio by 0.1-0.5 seconds can sometimes produce accidental alignments between visual impacts and the shifted audio.
>
> As we show in several examples on our project page: [C. TheGreatestHits Delayed Samples](https://anonymous-submission12338nci.github.io/random-random-supplementary-web/), a video shifted by 0.5 seconds can, in fact, appear more synchronized than one shifted by only 0.1 seconds.
>
> In contrast, datasets such as AVSync15, which include clear, non-rhythmic cues (e.g., crying, chicken crowing), exhibit the expected monotonic degradation under delay perturbations.
>
> Across all datasets (including TheGreatestHits), CycleSync consistently assigns *low scores when synchronization is absent*, which is the core requirement for a synchronization metric.
>
> We clarify this behavior in the revised appendix and include visual examples to illustrate how rhythmic repetition can create temporary alignment under small delays.
>
> We further clarify this in the revision.
>
> ## **(5) 5-seconds videos**
>
> We have included the 5-second videos on the project page: [A. 5-Second Generated Video Samples](https://anonymous-submission12338nci.github.io/random-random-supplementary-web/).
>
> We sincerely appreciate the reviewer’s continued engagement and believe the revision addresses all concerns comprehensively.

---

### Official Review · Reviewer_DdqZ · 2025-10-31

**Soundness:** 3
**Presentation:** 2
**Contribution:** 3
**Rating:** 4
**Confidence:** 4

**Summary:**

This paper proposes Syncphony, a diffusion-transformer-based framework for audio-to-video (A2V) generation that achieves fine-grained synchronization between sound and motion. The method introduces two key innovations: 1) Motion-aware Loss — reweights reconstruction loss to emphasize high-motion regions, improving temporal alignment. 2) Audio Sync Guidance (ASG) — uses an auxiliary “off-sync” model (without audio layers) to guide the full model during sampling toward stronger audio-motion coupling. To evaluate synchronization, the authors further introduce CycleSync, a new video-to-audio-based metric that measures whether the generated motion contains sufficient cues to reconstruct the original audio. Experiments on AVSync15 and TheGreatestHits datasets show improved synchronization and comparable or better visual quality relative to baselines such as AVSyncD and TempoTokens.

**Strengths:**

1. The paper proposes Motion-aware Loss and Audio Sync Guidance to improve the audio-visual synchronization of A2V generation, which are conceptually simple and empirically effective.
2. The proposed metric CycleSync offers a meaningful step forward over prior metrics (AV-Align, AlignSync, RelSync) by enabling evaluation at 24 fps and better correlating with human perception.
3. The paper also proposes a principled strategy to adapt a pretrained I2V model for the A2V task by selecting the most relevant layers to inject the audio cross-attention layer.
4. Syncphony consistently outperforms baselines on synchronization (CycleSync) and achieves competitive or superior FID/FVD, indicating that temporal precision does not come at the expense of visual quality.

**Weaknesses:**

1. While the paper’s ideas are sound, the architectural novelty is moderate. Syncphony heavily relies on a pretrained Pyramid Flow backbone, and the main innovations are at the loss and sampling levels rather than core model design.
2. The writing of the paper can be improved. For example, in the introduction, more details/motivations about the proposed methods could be included instead of the background information.
3. Limited baselines: the proposed method is only compared with AVSyncD and Pyramid Flow. Is it possible to include more baselines, such as those listed in AVSyncD?

**Questions:**

1. Fig 5 and 6: it will be helpful to include the ground-truth video as a reference.
2. How does ASG compare with the vanilla classifier-free guidance? For example, use the features with and without audio input as guidance.
3. The paper only finetunes the last 16 blocks (8–23) of the Pyramid Flow backbone. Are there any experimental results supporting the benefits of this choice, e.g., compared to finetuning the full model?
Others: see the weakness section.

---

> ### Author Response · Authors · 2025-11-21
>
> We sincerely thank the reviewer for recognizing the key strengths of our work, including the effectiveness of our Motion-aware Loss and Audio Sync Guidance, the advancement offered by CycleSync, the principled adaptation of the pretrained I2V model, and the strong synchronization and visual quality of Syncphony. We appreciate the thoughtful evaluation.
>
> Below, we provide a detailed response that addresses the reviewer’s concerns.
>
> ### (W1) Design Intent and Backbone Flexibility
>
> We would like to clarify (1) our design intent and (2) that the proposed methods are not tied to the Pyramid Flow backbone.
>
> **(1) Design intent**
>
> As we kindly clarify in lines L017, L053, and L302, our goal is not to propose a new video-generation backbone, but to introduce mechanisms for audio-visual synchronization on top of existing pretrained video models.
>
> Importantly, leveraging a pretrained backbone is not a lightweight modification. Training a video diffusion transformer from scratch is extremely resource-intensive, and naively fine-tuning the entire backbone degrades the pretrained model’s spatial and temporal generation quality. Therefore, the challenge is to introduce audio-motion synchronization without compromising the strong visual fidelity and temporal coherence already learned by the backbone. Syncphony is designed to address this challenge.
>
> **(2) The proposed methods are not tied to the Pyramid Flow backbone**
>
> In addition, we further emphasize that the proposed Motion-Aware Loss and Audio Sync Guidance, designed for audio-visual synchronization on pretrained backbones, are not tied to Pyramid Flow. We believe there is no architectural constraint that would prevent these mechanisms from being applied to other pretrained video models.
>
> Motion-Aware Loss computes motion magnitude from ground-truth consecutive frames and uses it to reweight the training objective, enabling the model to focus more on motion-relevant regions. This formulation is not tied to Pyramid Flow’s autoregressive structure; it could be applied equally to any diffusion/flow based video models, by weighting each frame’s prediction error using its corresponding ground-truth motion magnitude.
>
> Similarly, Audio Sync Guidance is designed for video models where visual generation and audio-conditioning pathways are functionally separated. In such architectures, ASG offers a clean and stable way to modulate the strength of audio cross-attention, and thus may extend beyond Pyramid Flow.
> In addition, as described in Appendix B.1, conventional classifier-free guidance (CFG) is trained by randomly dropping conditioning inputs as zero. However, we observe that directly applying this random-drop strategy to audio conditioning leads to degraded performance.
> Audio carries a meaningful semantic interpretation even when its value is dropped; silence is itself an informative condition. To address this issue, instead of dropping audio conditions, we propose an Audio Sync Guidance.
>
> In summary, our contributions do not depend on Pyramid Flow, and the proposed mechanisms may be applicable to other pretrained video models as well.
>
> ### (W2) Introduction Revision for Improved Method Context
> We have revised the introduction; please see lines 57–77. Here, we provide paragraphs of the revised introduction.
>
> > Despite these advancements, audio-to-video generation still faces a fundamental challenge: MSE-based objectives alone are insufficient for modeling accurate motion timing and appropriate motion magnitude. In diffusion or flow models, MSE is effective for reconstructing overall visual appearance, but it provides weak supervision for localized, precisely timed motion changes. Therefore, stronger and more targeted supervision is needed in regions that exhibit significant motion.
>
> > For training, we introduce a motion-focused loss that places greater emphasis on regions exhibiting significant visual movement. This encourages the model to better detect motion that is causally linked to the audio signal, even when such motion is highly localized.
> Furthermore, for sampling, we introduce a novel synchronization guidance strategy that enhances audio-driven motion without compromising visual fidelity. Motivated by the observation that applying traditional classifier-free guidance to audio conditions makes it difficult to train scenes without audio, we instead propose a guidance method that skips the audio layer itself, rather than dropping the audio condition.
>
> ### (Q1) Updated Figures with Ground truth
> We have updated **Figures 5 and 6** to include the ground truth reference videos. Thank you for the suggestion.

---

> ### Author Response · Authors · 2025-11-21
>
> ### (W3) Additional Baseline Comparisons
> Most prior audio-to-video models are either outdated, exhibit limited performance, or lack open-source implementations, which unfortunately makes it difficult for us to include them as direct baselines.
>
> Following AVSyncD, we reported TempoTokens in Table 1. However, as in AVSyncD, we could not include TempoTokens in Table 2 because the model is not trained on the TheGreatestHits dataset and therefore cannot produce comparable results.
>
> Also in line with AVSyncD, we will include TPoS and CoDi in our comparison. These additional baselines are currently being computed and will be updated once the evaluation completes.
>
> ### (Q2) ASG vs. Classifier-Free Guidance
> Conventional classifier-free guidance (CFG) in A2V is typically trained by randomly dropping conditioning inputs, a strategy that has proven highly effective in models such as text-to-image and class-conditional image generation. However, we observe that directly applying this random-drop strategy to audio conditioning leads to degraded performance.
>
> Unlike text, audio carries a meaningful semantic interpretation even when its value is *zero*. In other words, *silence* is itself an informative condition. When audio inputs are randomly dropped during training, the model is exposed to ambiguous signals where “absence of audio” and “silence as a semantic cue” become indistinguishable.
>
> Recent work, such as [ICG/TSG](https://openreview.net/forum?id=b3CzCCCILJ), also reports that treating the null state differently across timesteps can yield better results than CFG.
>
> In line with this motivation, we propose an Audio Sync Guidance that skips audio-conditioning layers during inference rather than dropping audio conditions, which is more suitable for A2V models.
>
> We added it to Appendix B.1. Thank you for your question.
>
> ### (Q3) Finetuning Later Transformer Blocks
> During our initial Image-to-Video training on the AVSync15 dataset (without Motion-Aware Loss and Audio Sync Guidance), we observed that finetuning the full Pyramid Flow backbone led to increased spatio-temporal artifacts and overall instability in the generated videos.
>
> In contrast, finetuning only the later blocks preserved the pretrained appearance priors, produced motion that aligned better with the text prompt, and improved both visual and temporal coherence.
>
> These observations are consistent with the structural characteristics of the backbone described in Appendix D.1.2; the earlier blocks primarily govern semantic appearance, while the later blocks are responsible for motion generation and temporal refinement.
>
> Since our goal is to introduce audio-synchronized motion while preserving the pretrained visual quality, only finetuning the motion-related later blocks is a more appropriate choice.
>
> Furthermore, with a small audio-visual dataset such as AVSync15, full-model finetuning easily leads to overfitting and appearance drift, whereas training only the motion-sensitive blocks yields more stable and efficient behavior.

---

> ### Author Response · Authors · 2025-12-03
>
> We sincerely thank the reviewer and the AC for their careful assessment of our work. Below, we briefly summarize how we have addressed the main concerns about backbone reliance, baselines, guidance design, and metric robustness.
>
> **(1) Design intent and backbone reliance**
>
> Reviewer DdqZ commented that Syncphony heavily relies on a pretrained Pyramid Flow backbone.
>
> Our goal is not to propose a new video generation backbone, but to provide a general method for **audio-motion synchronization on top of strong pretrained video models**. Syncphony is therefore designed to (i) preserve the visual quality and temporal coherence of a pretrained backbone, while (ii) adding mechanisms that capture fine-grained audio-motion relationships.
>
> To further clarify that Syncphony is not a “Pyramid Flow-only” method, we ran additional experiments on two more backbones, WAN-2.2 S2V and ASVA, in project page: [B. Application of ASG to Other A2V Model](https://anonymous-submission12338nci.github.io/random-random-supplementary-web). When we replace conventional classifier-free guidance (CFG) with our Audio Sync Guidance (ASG), we consistently observe *stronger lip motion and clearer audio-dependent temporal cues*, even though these backbones are architecturally different from Pyramid Flow (in [B. Application of ASG to Other A2V Model](https://anonymous-submission12338nci.github.io/random-random-supplementary-web)), supporting our claim that ASG behaves architecture-agnostically rather than being a one-off trick for a specific model. Please refer to our [anonymized project page](https://anonymous-submission12338nci.github.io/random-random-supplementary-web) for more details.
>
> **(2) Additional baselines and efficiency**
>
> Following the reviewer’s suggestion and the setup in AVSyncD, we extended our comparisons beyond TempoTokens to also include CoDi [1] and TPoS [2]. We will update this table clearly before the camera-ready version.
>
> While these methods are somewhat older and achieve clearly weaker synchronization and/or visual quality than recent approaches, we believe they still provide useful context for comparison.
>
> **Table: Quantitative results on the AVSync15 dataset.**
>
> | **Input** | **Model** | **FID ↓** | **FVD ↓** | **IA ↑** | **IT ↑** |
> | --- | --- | --- | --- | --- | --- |
> | **T+A** | TempoTokens | 8.9 | 4187.2 | 27.24 | 27.88 |
> | **I+T** | Pyramid Flow | 8.9 | 550.7 | – | 29.34 |
> |  | Pyramid Flow (fine-tuned) | 8.5 | 294.6 | – | 30.02 |
> | **I+T+A** | CoDi [1] | 14.5 | 1522.6 | 28.15 | 23.42 |
> |  | TPoS [2] | 11.9 | 1227.8 | 38.36 | 30.73 |
> |  | AVSyncD | 9.2 | 491.5 | 35.23 | 30.18 |
> |  | **Ours (Syncphony)** | 8.5 | 293.1 | 37.02 | 30.23 |
> | **Groundtruth** | – | – | – | 37.06 | 30.18 |
>
> [1] CoDi: Any-to-Any Generation via Composable Diffusion
>
> [2] The Power of Sound (TPoS): Audio Reactive Video Generation with Stable Diffusion
>
> **(3) ASG vs. CFG: practical effect and conceptual motivation**
>
> Existing A2V systems, such as ASVA, SUASVA, and even WAN-2.2 S2V, implement CFG by dropping audio inputs to zero, effectively treating silence as the “unconditional signal.” Our clarification during the discussion period was not meant to claim that CFG itself is flawed, but rather that this particular implementation is mismatched to the semantics of audio: zero-valued audio corresponds to meaningful silence, not to “no condition.”
>
> We designed Audio Sync Guidance (ASG) as a new guidance strategy that constructs the “without-audio” branch by skipping audio-conditioning layers, avoiding the semantic ambiguity of silence. Our additional experiments on Pyramid Flow, WAN-2.2 S2V, and ASVA consistently show that **ASG yields stronger audio responsiveness than conventional zero-audio CFG** in practice (especially in lip motion and audio-dependent temporal structure); project page: [B. Application of ASG to Other A2V Model](https://anonymous-submission12338nci.github.io/random-random-supplementary-web/).
>
> We hope this clarifies that the guidance design itself constitutes a meaningful and practically useful contribution for A2V models.

---

### Official Review · Reviewer_4rJg · 2025-11-01

**Soundness:** 2
**Presentation:** 3
**Contribution:** 3
**Rating:** 4
**Confidence:** 4

**Summary:**

Syncphony demonstrates meaningful progress in the audio-to-video generation domain. Its main contributions include: (1) a motion-aware loss that re-weights the training objective based on optical flow to focus the model’s learning on motion-intensive regions; (2) a training-free guidance technique that enhances the injection of audio information during inference without additional training; and (3) a more intuitive and reasonable evaluation metric for measuring audio-visual synchronization through reconstructed audio from generated videos. Experiments on two public datasets and qualitative case studies further validate the method’s effectiveness in improving both synchronization and visual quality.

**Strengths:**

- The proposed training-free guidance is novel and effectively points out the core challenge of achieving precise spatiotemporal alignment in audio-to-video generation.

- The proposed evaluation metric is intuitively reasonable and addresses the limitation of conventional metrics like FVD, which fail to effectively measure spatiotemporal alignment.

- The use of RoPE for spatiotemporal positional encoding enhances temporal consistency and spatial coherence in video generation, contributing to smoother and more structured motion representation.

- The demo results demonstrate good temporal alignment consistency.

**Weaknesses:**

Although the work is technically sound, there are several issues that I feel must be discussed.

- Regarding the motion-aware loss, it relies on a strong assumption that the visual content remains static and confined to a single scene. This assumption holds almost perfectly under the authors’ setting where short video clips of around two seconds. However, in more realistic video generation scenarios, background changes, camera motion, and scene transitions can occur without producing strong audio cues, but may catastrophically distort optical flow estimation and thus limit the scalability of the proposed approach.

- The proposed guidance technique randomly drops audio cross-attention layers during inference; however, this inference structure (unlike CFG or Autoguidance) is never encountered during training, making the resulting “unconditional” outputs less predictable. Moreover, although the authors provide some visual demonstrations, the underlying motivation may primarily apply to relatively smaller models where audio conditioning tends to be weaker. In larger-scale video generation frameworks such as Wan or Hunyuan-Video, audio conditions may not be as easily ignored, and models could exhibit different skip-layer behaviors. Rather than validating only on additional datasets, I would prefer the authors to verify this idea across multiple video generation baselines to strengthen the generality of their claims.

- Although the proposed new metric is intuitively reasonable, current video-to-audio (V2A) models also suffer from (or are still addressing) spatiotemporal alignment issues, which means they may not serve as a fully reliable ground-truth proxy. Moreover, the approach fundamentally increases evaluation time, potentially limiting scalability to larger experiments.

**Questions:**

- Could the authors elaborate on how they plan to address the challenges posed by more realistic (or longer) audio-to-video scenarios, where optical flow estimation can be severely affected by background motion, camera movement, or scene transitions beyond the main subject?

- Could the authors show how the skip-layer behavior manifests in other video generation models and whether similar phenomena can be consistently observed? In addition, if temporal alignment is indeed a crucial aspect of the task, why not consider using energy-based audio features as a more direct form of control or guidance?

- Given that modern multimodal large language models (e.g., Gemini 2.5) already demonstrate strong capabilities in understanding audio-visual information, and that existing video-to-audio alignment metrics (such as DeSync, which can be applied similarity in a2v field) provide reliable proxies for spatiotemporal correspondence, could the authors clarify or compare what specific advantages their proposed metric offers over these established approaches?

---

> ### Author Response · Authors · 2025-11-21
> **Response to 4rJg**
>
> We thank the reviewer 4rJg for the thoughtful evaluation and for highlighting several key strengths of our work. In particular, we appreciate the recognition of (1) the novelty of our training-free guidance for precise audio-spatiotemporal alignment, (2) the intuitive value of our proposed evaluation metric in addressing the limitations of conventional measures, (3) the effectiveness of RoPE in contributing to smoother and more structured motion representation, and (4) the strong temporal alignment shown in our demo results. We are grateful that these contributions were clearly acknowledged, as they represent the core motivations behind our design.
>
> Below, we provide a detailed response that addresses the reviewer’s concerns.
>
> ### (W1, Q1) Motion-Aware Loss in Dynamic Video Scenarios
> We would first like to clarify that our Motion-aware Loss *does not rely on optical flow*. As described in line 239, the loss is computed using the magnitude of ground-truth motion between consecutive frames, which directly reflects motion intensity without any optical-flow estimation. Because of this design, it remains robust to background changes, camera movement, and scene transitions. We therefore believe that the scalability concerns raised by the reviewer are less applicable in the context of our method.
>
> Moreover, Motion-aware Loss does not assume static backgrounds or single-scene settings. Its purpose is to amplify the learning signal in regions where motion occurs, guiding the model to focus more on the dynamic parts of the video. At that time, the model concentrates on these motion regions, and it learns why the motion happens. If the motion is caused by audio, the model learns to associate it with the audio; if the motion is unrelated to audio (e.g., background motion, camera movement), the model learns to treat it as sound-unrelated motion. In this way, the model generates synchronized sound-related motion appropriately while still generating sound-unrelated motion.
>
> The training dataset we used includes various camera motions such as zoom-in/zoom-out. Our generated results preserve these sound-unrelated motions while synchronizing only the sound-related parts. For example, in the “dog barking” sample in the Comparison section of our demo, the dog barks in sync with the audio while the camera continues to zoom, illustrating the model’s ability to distinguish sound-driven from non-sound-driven motion.
>
> In summary, Motion-aware Loss does not rely on static-scene assumptions and is robust to dynamic content. It strengthens the learning of audio-motion relationships regardless of video length or the presence of background or camera motion. We will clarify these points in the revised version.
>
>
> ### (W2, Q2-1) ASG Behavior and Scalability to Larger Backbones
> Thank you for the insightful question. To restate your point more clearly, we compute the guidance score of the model with all audio layers dropped.
>
> In Appendix B.2, we quantitatively show that the visual content remains stable when all audio layers are skipped. We kindly refer the reviewer to Table 4 for the corresponding experimental results.
>
> Regarding the reviewer’s concern about scaling to larger video-generation models, all experiments in our submission were conducted on four RTX 3090 GPUs, which limited our ability to test on large-scale backbones such as WAN. We agree that validating skip-layer behavior across multiple backbones would strengthen the generality of our claims. For this reason, we have started additional experiments on the WAN backbone to check whether skipping audio layers preserves video fidelity and whether ASG behaves consistently in this larger model.
>
> *We will report the results within the rebuttal period.*

---

> ### Author Response · Authors · 2025-11-21
>
> ### (Q2-2) Energy-Based Features and Temporal Alignment
> Thank you for the question. Temporal alignment is indeed a central challenge in audio-to-video generation, but as noted in line 245, the motion to be generated does not always exhibit a one-to-one temporal correspondence with the audio signal.
>
> As described in Appendix L613,
>
> > audio events are often accompanied by diverse motion patterns that vary by context.
> >
> >
> > For example, some events, such as a gunshot or a dog’s bark, occur with motion and sound nearly perfectly aligned. However, many others do not. A lion may start moving its mouth and body before emitting a roar. A person winds up before throwing a ball. A hammering motion may start before the sound and end just as the impact occurs. A trombone player moves the instrument before the sound begins. A bowling ball rolls with a low rumble before producing a sharp impact sound when it hits the pins.
> >
> > Therefore, synchronizing motion to audio does not mean matching peak amplitudes. Rather, it involves capturing the causal and contextual patterns of motion that correspond to different types of audio events.
> >
>
> Because of these diverse and context-dependent timing structures, using energy-based audio features as a direct control signal risks oversimplifying the audio-motion relationship and may introduce artifacts or harm generalization.
>
> Instead, our method is built on three complementary components that collectively facilitate contextual and non-rigid temporal alignment:
>
> - Audio cross-attention layers learn the semantic-temporal structure of the audio.
> - Motion-Aware Loss enhances learning signals in motion-relevant regions, complementing the audio conditioning mechanism.
> - Audio Sync Guidance further amplifies the influence of audio cross-attention, reinforcing natural, context-aware alignment.
>
> This design guides the model toward natural, context-dependent temporal alignment without imposing rigid timing assumptions, and it generalizes well across diverse audio-motion dynamics.
>
> ### (W3-1) Reliability of CycleSync
> As reviewer DdqZ mentioned, CycleSync offers a meaningful step forward over prior metrics (AV-Align, AlignSync, RelSync) by better correlating with human perception.
> CycleSync introduces a novel evaluation perspective by measuring whether the generated video contains sufficient timing and motion cues to reproduce the original audio structure (line 323), a capability not provided by existing metrics. As shown in line 431, CycleSync achieves the highest correlation with human judgments among all metrics we compare against.
>
> As discussed in Appendix C.4 (line 858), existing metrics often favor models that produce highly dynamic motion with plausible timing, even when the motion is not actually aligned with the audio signal. In contrast, CycleSync explicitly penalizes such audio-motion mismatches, allowing it to detect misalignment cases that existing metrics couldn't.
>
> We acknowledge, as noted in line 485 and Appendix C.5, that current V2A models may still exhibit spatiotemporal alignment issues. However, our experiments show that even with today’s V2A model, CycleSync is already able to capture misalignment patterns that other metrics fail to recognize. Moreover, as V2A models continue to improve, we expect CycleSync’s reliability and discriminative power to further increase.
>
> ### (W3-2) Evaluation Runtime of CycleSync
> Regarding evaluation time, V2A inference takes approximately 8 seconds per sample with a batch size of 1, and evaluating the entire AVSync dataset (432 samples) required about 1 hour. Given that CycleSync captures audio-motion misalignments that existing metrics miss, we believe this additional cost represents a reasonable trade-off for improved reliability. Moreover, V2A inference can be efficiently parallelized through batching and multi-GPU processing, so the computational overhead does not become a bottleneck even in larger-scale evaluations.

---

> ### Author Response · Authors · 2025-11-21
>
> ### (Q3) CycleSync Compared with Existing Alignment Metrics
>
> Thank you for the question. The key advantage of CycleSync is its ability to capture context-dependent audio-motion alignment, which existing metrics fail to detect. From our empirical comparisons, DeSync performs well on clean, clearly aligned sync events but struggles with motions that include preparatory, residual, or context-driven movement, such as the rebound of a drumstick or sweeping motions where sound does not occur at every movement peak.
>
> We extended the human alignment study in Appendix C.3 to include DeSync. The setup was:
>
> - 20 videos sampled from Pyramid Flow (audio-unrelated motion) and Syncphony (audio-related motion),
> - 9 human participants rating sync quality on a 1-5 scale,
> - Pearson correlations computed between human scores and metric outputs.
>
> **Model Ranking Agreement (Appendix C.3 Table 8):**
>
> | Model | Human Score ↑ | CycleSync ↑ | AV-Align ↑ | RekSync ↑ | DeSync ↓ |
> | --- | --- | --- | --- | --- | --- |
> | Pyramid Flow (I2V) | 2.68 | 8.15 | **24.96** | **55.36** | 0.88 |
> | Syncphony (Ours) | **4.30** | **22.04** | 21.88 | 50.44 | **0.86** |
>
> **Correlation with Human Ratings (Appendix C.3 Table 7):**
>
> | Metric | Correlation | 95% CI Lower | 95% CI Upper |
> | --- | --- | --- | --- |
> | Human Score | 1.000 | 1.000 | 1.000 |
> | CycleSync | **0.486** | **0.053** | **0.919** |
> | DeSync | 0.206 | -0.279 | 0.690 |
> | AV-Align | 0.043 | -0.451 | 0.538 |
> | RelSync | -0.623 | -1.011 | -0.236 |
>
> As the tables show, while DeSync provides meaningful signals compared to prior metrics, CycleSync still exhibits the strongest alignment with human perception.
>
> Regarding multimodal LLMs such as Gemini 2.5: to the best of our knowledge, existing MLLMs can understand semantic audio-visual relationships but do not provide a quantitative synchronization metric that captures the fine-grained timing needed for A2V evaluation. If such work exists, we would appreciate being informed.
>
> We have updated it in Appendix C.3, from line 828 including Table 7 and 8.

---

> > ### Comment · Reviewer_4rJg · 2025-11-24
> >
> > Thanks for your reply.
> > My concerns about the motion-aware loss have been addressed, and I am willing to raise my score if the following questions can be further clarified:
> >
> > (1) In your reply to Q2-2, you used the examples of a lion running and throwing a ball, and argued that the method would oversimplify the audio–motion relationship. However, I still do not fully understand why the “motion anticipation” you mentioned should be related to the audio cross-attention layer. You argued that guiding the audio cross-attention layer could promote the synthesis of this component, because audio provides global contextual information. Yet audio’s global structure mainly reflects the acoustic scene and, intuitively, should not contain such highly causal world-knowledge about motion.
> >
> > (2) Regarding CycleSync, you used peak-curve alignment to measure model performance, which seems to contradict your rebuttal claim that “synchronizing motion to audio does not mean matching peak amplitudes.” If CycleSync becomes a future a2v benchmark, researchers could intentionally optimize for peak-amplitude alignment to score higher, which—as you argued—may instead introduce artifacts or harm generalization.

---

> > > ### Comment · Reviewer_4rJg · 2025-11-24
> > >
> > > BTW, if computational resources are limited, reproducing the Wan experiments does not need to be a priority at rebuttal period. Still, if there is an opportunity, showing that this phenomenon also appears in modern, more advanced generative models rather than only in smaller models would make the claim more broadly applicable.

---

> > > ### Author Response · Authors · 2025-11-25
> > >
> > > (1)
> > >
> > > Thank you for raising this point. We agree that audio does not contain explicit causal world-knowledge about motion. Our intention was not to claim that audio encodes these anticipatory motion patterns. Rather, our argument concerns how the audio-conditioned layers interpret audio cues as signals for motion timing for modulating when the video model’s motion prior should be activated relative to an audio event.
> > >
> > > When we stated that “audio cross-attention layers learn the semantic–temporal structure of the audio,” our meaning follows Section 3.3 (L261): the audio layers learn to extract semantic and temporal information and translate them into cues that modulate when the video model’s motion prior should be expressed.
> > >
> > > > In audio-conditioned video generation, audio-driven layers are responsible for injecting timing cues into the visual dynamics.
> > > >
> > >
> > > In summary, our statement does not imply that audio contains the anticipatory motion itself. Instead, the audio layers interpret semantic and temporal cues of audio to decide when motion should be activated in a way that is semantically and temporally aligned with the audio event.
> > >
> > > ---
> > >
> > > (2)
> > >
> > > Thank you for raising this concern. We would first like to clarify that our statement “synchronizing motion to audio does not mean matching peak amplitudes” means that alignment should not be evaluated by directly matching the peaks of the motion to the peaks of the audio. CycleSync does not compare motion peaks to audio peaks. Instead, it compares the original audio with the audio reconstructed from the generated video.
> > >
> > > Regarding the concern that researchers may intentionally optimize for peak-amplitude alignment to achieve higher CycleSync scores, such an approach would be computationally almost impossible.
> > >
> > > For the reconstructed audio to align with the original audio, the generated motion must contain sufficient timing and motion cues for a V2A model to reconstruct the temporal structure of the original audio.
> > >
> > > Thus, because CycleSync inherently encourages capturing temporally structured motion through a V2A model rather than simplistic audio-motion peak matching, forcing the reconstructed audio peaks to align with the original audio peaks would be practically impossible.
> > >
> > > In short, CycleSync does not compare motion peaks to audio peaks; rather, it compares the original audio with the audio reconstructed from the generated video. Because it is practically impossible to intentionally optimize for aligning reconstructed and original audio peaks, the concern about peak-based optimization does not apply.
> > >
> > > ---
> > >
> > > If our responses have resolved your concerns, we would be grateful if you could consider adjusting your score accordingly.

---

> > > > ### Comment · Reviewer_4rJg · 2025-11-26
> > > >
> > > > Thank you for the response. I think I still need to clarify several points. First, according to L334–340, the CycleSync score compares the onset peaks between two audio signals, which are directly tied to the absolute energy of the audio and are used as a proxy for the generated audio’s temporal structure. However, this metric does not capture motion anticipation in the generated video (which you explicitly argue in the rebuttal that this information cannot be provided by energy-like temporal guidance). Following the logic of the metric, a video that is more tightly aligned to the audio’s temporal structure naturally receives a higher score. This implies that a video-generation model that simply treats the audio’s temporal structure as guidance or a conditioning signal can trivially produce audio-aligned temporal patterns and thus achieve a higher CycleSync score, which may also opens the door to potential metric hacking.

---

> > > > > ### Author Response · Authors · 2025-12-03
> > > > >
> > > > > Thank you for your thoughtful question. Since the discussion section is no longer available due to the OpenReview issue, we unfortunately cannot clarify interactively, but we will provide as complete an explanation as possible.
> > > > >
> > > > > **(1) Clarifying what CycleSync actually compares**
> > > > >
> > > > > As you correctly summarized, CycleSync computes the temporal agreement between the onset peaks of two audio signals $a$ and $\hat{a}$. What matters here is how these two audio signals are obtained:
> > > > >
> > > > > 1. Given an input audio $a$, the A2V model generates a video $v$.
> > > > > 2. Given the generated video $v$, a V2A model generates audio $\hat{a}$.
> > > > > 3. CycleSync then compares the onset peaks of $a$ and $\hat{a}$.
> > > > >
> > > > > Thus, CycleSync *does not* compare audio peaks to video motion peaks.
> > > > >
> > > > > This is important because the argument in our rebuttal “*that motion anticipation cannot be inferred solely from energy-based audio cues”* referred specifically to audio-video peak comparison. That is the limitation of metrics like AV-Align, which attempt to align audio peaks with visual peaks.
> > > > >
> > > > > CycleSync avoids this entirely by aligning audio-audio peaks, not audio-video.
> > > > >
> > > > > **(2) Why audio-video peak alignment is problematic**
> > > > >
> > > > > As we described in the paper and rebuttal, the temporal peak of audio does not directly match the temporal peak of visual motion:
> > > > >
> > > > > - In a *hammering* scene, the audio peak occurs *when* the hammer hits the object, but the motion visually slows or stops at that moment.
> > > > > - In a *sweeping* scene, the audio peak occurs *when* the sweeping motion is *fastest*.
> > > > >
> > > > > So audio and motion peaks do not have a direct 1-to-1 mapping relationship; it depends on the motion semantics.
> > > > >
> > > > > Metrics such as RelSync and DeSync try to learn this semantic-dependent alignment, but as shown in our experiments on the TheGreatestHits dataset, we observed that their scores can become unreliable, sometimes assigning higher values to videos that are not meaningfully related to the audio.
> > > > >
> > > > > (We have included details in the appendix.)
> > > > >
> > > > > **(3) Why CycleSync works better**
> > > > >
> > > > > CycleSync relies on a V2A model that we assume a V2A model can produce *semantic-temporally aligned* audio from the given video:
> > > > >
> > > > > - A hammering video should produce audio whose peaks occur at the hammer impact moments.
> > > > > - A sweeping video should produce audio whose peaks follow the sweeping rhythm.
> > > > >
> > > > > If the A2V model produces a video that is semantically and temporally consistent with the input audio, then the V2A model will naturally produce an audio signal $\hat{a}$ with a similar temporal peak structure. CycleSync measures exactly this.
> > > > >
> > > > > This is why CycleSync correlates strongly with human perception (as shown in our results) and therefore, it outperforms other metrics in capturing perceptual synchronization.
> > > > >
> > > > > **(4) On the possibility of “metric hacking”**
> > > > >
> > > > > Your concern is that “a model could simply follow the temporal structure of the conditioning audio and obtain a high score.”
> > > > >
> > > > > This is in fact **precisely what an audio-to-video model should do**, if the semantics match.
> > > > >
> > > > > If a model can generate videos whose V2A generated audio exhibits the correct semantic dependent peak structure (different for hammering, sweeping, etc.), then we would interpret that as successful A2V generation.
> > > > >
> > > > > Crucially, “hacking” CycleSync would require producing videos that yield semantically correct regenerated audio across diverse actions, each with different motion-audio relationships.
> > > > >
> > > > > That said, we acknowledge that, like any automatic metric, CycleSync is not theoretically immune to adversarial cases. No metric can fully eliminate this possibility. However, among existing metrics, CycleSync offers a substantially more robust and semantically grounded alignment signal, especially in scenarios where prior audio-video alignment metrics fail.
> > > > >
> > > > > CycleSync targets a specific property:
> > > > >
> > > > > *semantic audio-video temporal consistency*, measured through audio-audio alignment via V2A reconstruction.
> > > > >
> > > > > In practice, if one is concerned about *visual artifacts or unrealistic motion*, these issues are naturally captured by traditional metrics such as *FID/FVD*.
> > > > >
> > > > >
> > > > >
> > > > > **(5) Additional WAN experiments demonstrating ASG’s scalability to larger backbones**
> > > > >
> > > > > We also verified that our proposed ASG works well even on large models such as WAN. In WAN-2.2 (Speech-to-Video, 14B), ASG consistently improved lip-motion magnitude and audio-dependent temporal alignment compared to the conventional zero-audio CFG implementation.
> > > > >
> > > > > Please refer to the project page: [B. Application of ASG to Other A2V Model](https://anonymous-submission12338nci.github.io/random-random-supplementary-web/) for qualitative results.
> > > > >
> > > > > We hope this clarifies the motivation behind CycleSync and addresses your concerns regarding its behavior.
> > > > > As requested during the discussion, we also conducted additional WAN experiments, which demonstrate that ASG scales to larger backbones.
> > > > >
> > > > > Thank you again for raising this important point.

---

### Official Review · Reviewer_nWr3 · 2025-11-02

**Soundness:** 3
**Presentation:** 4
**Contribution:** 3
**Rating:** 6
**Confidence:** 3

**Summary:**

This paper introduces Syncphony, an audio-to-video (A2V) model capable of generating 380×640 resolution, 24fps videos synchronized with diverse audio inputs. Built upon a pre-trained video backbone, it emphasizes on audio-visual synchronization through two main contributions:
- Motion-aware Loss, a loss weighting mechanism emphasizing on the high-motion regions of the video.
- Audio Sync Guidance, a classifier-free guidance design focusing solely on audio conditioning to emphasize better synchronization.

Moreover the authors introduce an auxiliary contribution for the model evaluation:
- CycleSync, an audio similarity metrics between the detected peaks of the ground truth and the A2V -> (pretrained) V2A generated audio.

The A2V model is finetuned from a pretrained I2V model, by adding cross attention layers in the latter transformer blocks of the pretrained model. The backbone is a pretrained PyramidFlow Video model, trained on videos up to 5 seconds long at 24 fps and 380 × 640 resolution. Audio is sampled at 16kHz and encoded through DenseAV for conditioning. Text is encoded through CLIP. Temporally-aware RoPE frequencies are employed to aligned the modalities in the aforementioned cross attention layers.

Evaluations are conducted on the AVSync15 dataset using both objective (FID, FVD, Image-Audio similarity, Image-Text similarity, CycleSync) and subjective metrics (IQ, FC, Sync). They demonstrate that the proposed method outperforms the AVSyncD on all axes.

On the Greatest Hits dataset, the proposed method outperforms on the CycleSync and FVD metrics, showing its emphasis on both video generation quality and audio visual synchronization.

**Strengths:**

The paper quality is enforced by thorough ablations and experiments, presented both in the main sections and appendixes, notably:
- demonstration of the greater sensitivity of the CycleSync metrics compared with previously proposed ones, running correlation with human study and experimenting with controlled settings such as temporal shifts.
- ablations on the Motion-aware Loss and Audio Sync Guidance contributions
- pretrained model behavior study to understand where to inject the finetuning layers

Although this paper presents an Audio to Video model, its contributions should translate into better the Video to Audio model designs.

**Weaknesses:**

The Figure 1 is not clear enough to me. It is not clear what the frozen and trainable layers refer to (are those transformer blocks?). I would suggest adding in the captions that the audio features are injected in the latter blocks (the expectation is usually to add cross attention to each block so the presentation is quite counter intuitive without a corresponding explanation).

**Questions:**

Formatting:
- Add spacing before opening parentheses (around the line 347).
- In the table 3, I suggest better highlighting which model is the final version (maybe in the top row and by adding a row separator).
- As one of the paper's contributions is a classifier-free-guidance design, I would suggest adding some ablations of the choice of the cfg parameters.

---

> ### Author Response · Authors · 2025-11-21
> **Response to nWr3**
>
> We sincerely appreciate nWr3's insightful comments and suggestions.
>
> We appreciate that the reviewer has carefully recognized several core strengths of our work, including the comprehensive ablations, the demonstrated sensitivity of the CycleSync metrics, the analysis of pretrained model behavior, and the broader applicability of our contributions. We are grateful that these aspects were acknowledged, as they reflect key design principles that guided our approach.
>
> ### (W1) Clarifying Trainable Blocks and Audio Injection in Figure 1
> Thank you for pointing this out. In our method, when the transformer blocks are divided into two chunks, we insert the audio cross-attention layers only into the latter transformer blocks, directly before the joint self-attention, and train only these latter blocks. We agree that this was not clearly reflected in the caption. We have **updated the Figure 1** to explicitly state that audio features are injected only into the later blocks and that only these blocks are trainable. Thank you for the helpful suggestion.
>
> ### (Q1, Q2) Table and Formatting Improvements
> We have added a space before the opening parenthesis and **updated Table 3** by adding a row separator to clearly highlight the final model.
>
> ### (Q3) Ablation on Guidance Scales Included
> Table 3 includes an ablation over different guidance scales (0, 1, 2, and 4). Please let us know if you were referring to an additional type of analysis.

---

> > ### Comment · Reviewer_nWr3 · 2025-11-26
> >
> > I thank the authors for addressing my comments. In light of discussions with other reviewers, I am puzzled by the comments regarding ASG and its relation with silence, which I think tone down the significance of the contribution.
> >
> > I do not understand why classical CFG (e.g. randomly dropping conditioning inputs) leads to degraded performance. Applying CFG on audio should not translate into setting the conditioning waveform to zero (e.g. make it silent), but conditioning the model on some learnable classifier-free token or masking the cross attention to prevent the model from attending to the audio signal (while using zero-attention, thus translating into 0-valued attention output and 0-value FFN output if no FFN-bias, which would effectively equate to skipping the layer). In this work it sounds like the classical CFG implementation is faulty.

---

> > > ### Author Response · Authors · 2025-11-26
> > >
> > > First, we would like to clarify that the classical CFG the reviewer mentioned refers to the *CFG implementation that has actually been used in prior Audio-to-Video (A2V) works*. Existing A2V methods, including ASVA [1] and more recent approaches such as SUASVA [2], apply CFG by randomly dropping out the audio condition to zero. Our intention was to highlight and discuss the limitations in an intuitive way.
> > >
> > > (For reference: [ASVA audio-conditioning code](https://github.com/lzhangbj/ASVA/blob/658793825bca8c02586197d4638e4fcd7f1a3896/avgen/pipelines/pipeline_audio_cond_animation.py#L180))
> > >
> > > As you correctly pointed out, applying CFG to audio should not be equivalent to setting the conditioning waveform to zero. We fully agree with this. We also wanted to discuss about the mismatch between *how CFG is conceptually supposed to work* and *how it has been implemented in A2V pipelines to date*.
> > >
> > > Our proposed Audio Sync Guidance (ASG) is a new form of classifier-free guidance specifically designed for A2V generation, which has not been explored in prior work. It provides the signal for the "without-audio condition" that CFG is conceptually supposed to do, but has never been properly implemented in previous A2V works.
> > >
> > > Moreover, because attention projection layers include biases, using a silent (zero-valued) audio as an input inherently forces the model to interpret these biases as representing “silence,” rather than representing a dropped condition. For this reason, we argue that the same zero-drop strategy used in some of text-to-image or other domains is not semantically appropriate for audio.
> > >
> > > In contrast, our ASG provides *an intuitive and empirically effective guidance method*, offering a more suitable formulation for A2V models. We believe ASG represents a novel and generalizable guidance strategy, and we hope it can serve as a useful direction for future A2V research.
> > >
> > > [1] Audio-Synchronized Visual Animation, ECCV2024, https://arxiv.org/abs/2403.05659
> > >
> > > [2] Scaling Up Audio-Synchronized Visual Animation: An Efficient Training Paradigm, Preprint. Under review, https://arxiv.org/abs/2508.03955

---

> > > ### Author Response · Authors · 2025-12-03
> > >
> > > We sincerely thank Reviewer nWr3 for the continued discussion and the thoughtful questions regarding our method. Since the discussion section is no longer available due to the OpenReview issue, we would like to provide a brief, final clarification along with additional supporting results.
> > >
> > > **(1) Figure 1 and Formatting Revision**
> > >
> > > The reviewer pointed out Figure 1 and formatting improvements. We have revised our paper clearly.
> > >
> > > **(2) Clarification on CFG Ablation**
> > >
> > > As noted earlier, Table 3 already includes ablations over guidance scales.
> > >
> > > **(3) Clarification on ASG vs. Classical CFG**
> > >
> > > Regarding the reviewer’s concern about ASG and its relation to classical CFG, we would like to clarify that our intention was not to suggest that CFG is conceptually problematic. Rather, we observed that *the specific CFG implementations used in existing A2V pipelines* (e.g., ASVA [1], SUASVA [2], and WAN-2.2 S2V [3]) commonly rely on zero-audio dropout, and this practice does not faithfully represent “without-condition” for audio because zero-valued audio represents meaningful “silence”. ASG was introduced to provide a more principled alternative that aligns with the conceptual goal of CFG in the audio-conditioned setting.
> > >
> > > To further validate the effectiveness and generality of ASG, we additionally applied it to multiple video backbones beyond Pyramid Flow.
> > >
> > > - **WAN-2.2 (Speech-to-Video):** ASG consistently improved lip-motion magnitude and audio-dependent temporal alignment over classical CFG implementation.
> > > - **ASVA:** ASG produced clearer audio responsiveness across diverse inputs.
> > >
> > > Qualitative examples are available on our project page: [B. Application of ASG to Other A2V Model](https://anonymous-submission12338nci.github.io/random-random-supplementary-web/).
> > >
> > > Together, these results demonstrate that ASG offers a conceptually sound and empirically robust improvement over the conventional zero-audio CFG implementation in A2V, while retaining the conceptual goal of CFG and remaining simple and broadly applicable.
> > >
> > > [1] ASVA: Audio-Synchronized Visual Animation
> > >
> > > [2] SUASVA: Scaling Up Audio-Synchronized Visual Animation: An Efficient Training Paradigm
> > >
> > > [3] Wan: Open and Advanced Large-Scale Video Generative Models

---

### Author Response · Authors · 2025-11-21
**We thank the reviewers**

We thank the reviewers for their thoughtful feedback. Below, we address the common concerns raised across the reviews and clarify key points of our method.

## Design Intent and Backbone Flexibility
Reviewers DdqZ and 2TxV raised the concern that Syncphony heavily relies on the Pyramid Flow backbone with only moderate architectural extensions.

We would like to clarify (1) our design intent and (2) that the proposed methods are not tied to the Pyramid Flow backbone.

### (1) Design intent
As we kindly clarify in lines L017, L053, and L302, our goal is not to propose a new video-generation backbone, but to introduce mechanisms for audio-visual synchronization on top of existing pretrained video models.

Importantly, leveraging a pretrained backbone is not a lightweight modification. Training a video diffusion transformer from scratch is extremely resource-intensive, and naively fine-tuning the entire backbone degrades the pretrained model’s spatial and temporal generation quality. Therefore, the challenge is to introduce audio-motion synchronization without compromising the strong visual fidelity and temporal coherence already learned by the backbone. Syncphony is designed to address this challenge.

### (2) The proposed methods are not tied to the Pyramid Flow backbone
In addition, we further emphasize that the proposed Motion-Aware Loss and Audio Sync Guidance, designed for audio-visual synchronization on pretrained backbones, are not tied to Pyramid Flow. We believe there is no architectural constraint that would prevent these mechanisms from being applied to other pretrained video models.

Motion-Aware Loss computes motion magnitude from ground-truth consecutive frames and uses it to reweight the training objective, enabling the model to focus more on motion-relevant regions. This formulation is not tied to Pyramid Flow’s autoregressive structure; it could be applied equally to any diffusion/flow based video models, by weighting each frame’s prediction error using its corresponding ground-truth motion magnitude.

Similarly, Audio Sync Guidance is designed for video models where visual generation and audio-conditioning pathways are functionally separated. In such architectures, ASG offers a clean and stable way to modulate the strength of audio cross-attention, and thus may extend beyond Pyramid Flow.
In addition, as described in Appendix B.1, conventional classifier-free guidance (CFG) is trained by randomly dropping conditioning inputs as zero. However, we observe that directly applying this random-drop strategy to audio conditioning leads to degraded performance.
Audio carries a meaningful semantic interpretation even when its value is dropped; silence is itself an informative condition. To address this issue, instead of dropping audio conditions, we propose an Audio Sync Guidance.

In summary, our contributions do not depend on Pyramid Flow, and the proposed mechanisms may be applicable to other pretrained video models as well.

## Clarifying the Robustness of Motion-Aware Loss
Reviewers 4rJg and 2TxV raised concerns that Motion-Aware Loss assumes static scenes and that simple latent frame differences may not capture meaningful sound-related motion.

We would like to clarify that Motion-Aware Loss remains effective even in dynamic environments where motion is not semantically aligned with the audio (e.g., background motion, camera movement), and still guides the model to learn meaningful audio-motion relationships.

As described in line 239, Motion-Aware Loss computes the magnitude of ground-truth motion between consecutive frames and uses it to reweight the loss. This mechanism strengthens the learning signal specifically in motion regions, encouraging the model to focus on dynamic areas and generate them more accurately.

Through this process, the model learns the cause of each motion.

- If the motion is caused by audio, the model learns to associate it with the audio;
- if the motion is unrelated to audio (e.g., background motion, camera movement), the model learns to treat it as sound-unrelated motion.

In this way, the model generates synchronized sound-related motion appropriately while still generating sound-unrelated motion.

The training dataset we used includes various camera motions such as zoom-in/zoom-out. Our generated results preserve these sound-unrelated motions while synchronizing only the sound-related parts. For example, in the “dog barking” sample in the Comparison section of our demo, the dog barks in sync with the audio while the camera continues to zoom, illustrating the model’s ability to distinguish sound-driven from non-sound-driven motion.

In summary, Motion-Aware Loss effectively guide the model to learn meaningful audio-motion alignment even in dynamic, audio-unrelated motion environments.

---

### Author Response · Authors · 2025-12-03
**Issues raised by reviewers and our resolutions (1/2)**

Dear Area Chair,

We sincerely appreciate the Area Chairs’ efforts in maintaining the integrity of the review process.
Below we summarize how the discussion phase resolved the key concerns raised.

## Generality Beyond Pyramid Flow Backbone

### **Positive assessment.**

Reviewers nWr3 and DdqZ praised our principled and well-analyzed layer-selection strategy for adapting pretrained I2V backbones to A2V by identifying where audio cross-attention should be injected with our layer-skip experiments.

### **Concern.**

Reviewers DdqZ and 2TxV raised the concern that Syncphony heavily relies on the Pyramid Flow backbone with only moderate architectural extensions.

### **Resolution.**

We emphasized that our goal is to provide general mechanisms for *audio-synchronized* video generation on the *pretrained* video model. To demonstrate generality, we conducted new experiments on WAN-2.2 (Speech2Video) and ASVA, two architecturally different A2V models, during rebuttal.

Across both models, ASG consistently improved lip motion and temporal alignment, and we explained Motion-Aware Loss could be applied equally to diverse diffusion/flow-based video models. Reviewer nWr3 also mentioned *our contributions should translate into better Video to Audio model designs*. This suggests that our contributions, ASG and Motion-Aware Loss, are *architecture-agnostic* and resolve the backbone concern directly.

---

## Robustness of Motion-Aware Loss

### **Positive assessment.**

Reviewers (nWr3, DdqZ) positively noted that Motion-Aware Loss is simple, effective, and well-supported by ablations. They highlighted its empirical value in improving audio-visual synchronization.

### **Concern.**

Reviewers 4rJg and 2TxV raised concerns that Motion-Aware Loss assumes static scenes and that simple latent frame differences may not capture meaningful sound-related motion.

### **Resolution.**

We clarified that Motion-Aware Loss is effective even in dynamic scenes where motion is not semantically aligned with the audio, and still guides the model to learn meaningful audio-motion relationships.

Here, we provide the short explanation: it reweights the loss function based on ground-truth motion magnitude, enabling the model to naturally learn (1) audio-driven motion (if motion is aligned with sound), and (2) audio-independent motion (if not, such as camera/background movement).

We showed that synchronized motion remains accurate even under dynamic conditions (e.g., dog barking while the camera zooms).

Reviewer 4rJg stated that their concerns regarding Motion-Aware Loss were addressed and expressed willingness to raise the score. We believe our resolution addressed the concern fully.

---

## CycleSync Robustness

### **Positive assessment.**

All reviewers (nWr3, 4rJg, DdqZ, 2TxV) consistently praised that CycleSync is a reasonable and meaningful improvement over prior metrics, showing higher sensitivity and stronger correlation with human judgments.

### **Concern.**

Reviewers 4rJg and 2TxV raised the concern that current V2A models are imperfect, and thus may not provide a fully reliable basis for evaluation.

### **Resolution.**

CycleSync focuses on semantic and temporal consistency rather than low-level signal matching. Prior metrics (AV-Align, AlignSync, RelSync) cannot capture it, especially when the models generate highly dynamic motion with plausible timing, even when the motion is not actually aligned with the audio signal.

Even with today’s V2A models, CycleSync shows the strongest correlation with human evaluations among all compared metrics, and already identifies misalignment cases that prior metrics fail to detect, demonstrating practical reliability. Moreover, as V2A models continue to improve, we expect CycleSync’s reliability and discriminative power to increase further.

---

> ### Author Response · Authors · 2025-12-03
> **Issues raised by reviewers and our resolutions (2/2)**
>
> ## Clarification on ASG vs. Classical CFG
>
> ### **Positive assessment.**
>
> Reviewers (nWr3, 4rJg, DdqZ) highlighted that our Audio Sync Guidance (ASG) is conceptually simple, training-free, and effectively improves audio-visual synchronization, and that our ablations clearly validate its contribution.
>
> ### **Concern.**
>
> Reviewer nWr3 and 2TxV raised questions about ASG and its relation to classical CFG.
>
> ### **Our response.**
>
> We clarified that the classical CFG refers to the CFG implementation that has actually been used in prior Audio-to-Video (A2V) works. Existing A2V methods implement CFG by dropping audio inputs to zero, treating silence as the “unconditional signal.” Our clarification during the discussion period was not meant to claim that CFG itself is flawed, but rather that this particular implementation is mismatched to the semantics of audio: zero-valued audio corresponds to meaningful silence, not to “no condition.”
>
> ASG is introduced to provide a cleaner unconditional pathway by skipping audio-conditioning layers, avoiding the semantic ambiguity of silence. Our additional experiments on WAN-2.2 S2V and ASVA consistently show that **ASG yields stronger audio responsiveness than CFG** in practice (especially in lip motion and audio-dependent temporal structure); project page: [B. Application of ASG to Other A2V Model](https://anonymous-submission12338nci.github.io/random-random-supplementary-web/).
>
> In short, ASG offers a conceptually sound and empirically robust improvement over the conventional zero-audio CFG implementation in A2V, while retaining the core principles of CFG and remaining simple and broadly applicable.
>
> ---
>
> ## **Additional Clarifications & Improvements**
>
> During the discussion we also:
>
> - expanded human evaluation (more participants & videos),
> - added baselines (CoDi, TPoS),
> - explained temporal-shift behavior on TheGreatestHits dataset,
> - clarified the purpose of “Pyramid Flow (fine-tuned),”
> - demonstrated that Syncphony remains competitive in efficiency,
> - and extended analyses to address all reviewer questions.
>
> Two reviewers (nWr3, 4rJg) explicitly stated that their major concerns were almost resolved, and Reviewer 4rJg indicated willingness to raise the score.
>
> ---
>
> ## **Final Remark**
>
> Given the positive assessments, the added experiments, and the resolution of all major concerns, we respectfully believe the paper now leans toward acceptance.
>
> We hope the AC will heavily weigh the comprehensive nature of our response and the extensive new evidence provided, which the reviewers were unable to acknowledge due to the freeze. Thank you for your time and efforts.
>
> Best regards,
>
> The Authors

---

### Meta-Review · Area_Chair_R1ag · 2026-01-06

**Summary:**

Syncphony fine-tunes a pretrained Pyramid Flow video backbone for audio-to-video (380×640, 24fps) with better audio–motion synchronization.
- Reviewers focused on three contributions: motion-aware loss, Audio Sync Guidance (ASG), and the CycleSync V2A-based synchronization metric. The decision is anchored on whether motion loss is robust, whether ASG is meaningfully different from CFG practice in A2V and generalizes beyond Pyramid Flow, and whether CycleSync is reliable given V2A dependence and peak-based matching; efficiency and incremental contribution over Pyramid Flow were secondary.

**Reviewer Concerns:**

Concerns largely addressed
- motion loss clarified it does not rely on optical flow; uses magnitude of ground-truth motion between consecutive frames; reviewer stated the concern was addressed.
- Clarity of Fig.1 frozen/trainable blocks and audio injection clarified; table/formatting and adding ground-truth references in figures addressed.
- CycleSync human study scale, expanded evaluation (participants and videos) and updated correlations.

Outstanding concerns:
- ASG vs CFG (nWr3, 2TxV): reviewers argue CFG need not mean zero-audio; masking/learnable null could resemble skipping. Authors clarify they target zero-audio CFG as used in prior A2V and argue silence is meaningful; added qualitative evidence on WAN-2.2 S2V and ASVA, but limited direct quantitative comparison to a proper CFG baseline.
- CycleSync robustness (4rJg, 2TxV): concerns about peak-based alignment and metric hacking; authors clarify CycleSync is audio–audio via V2A reconstruction and argue hacking would require semantically correct regenerated audio across diverse actions; still a general limitation of automatic metrics.
- Generality beyond Pyramid Flow (4rJg, 2TxV): additional backbone evidence is mostly qualitative.
- Efficiency (2TxV): authors provide timing breakdown/normalization; practical usability concern remains.

**Reviewer Scores:**

- nWr3 likely stays 6
- 4rJg likely to increase to 6 ; motion-aware loss concern resolved, remaining CycleSync/ASG generality concerns partially addressed.
- DdqZ likely to increase to 6; figures, finetuning rationale, and added baselines addressed; novelty noted as moderate.
- 2TxV likely stays 4 (at most 4 to 6) given persistent concerns on single-backbone evidence, CFG comparison, and efficiency.

Based on these I recommend acceptance of the paper as a poster.

---

### Decision · Program_Chairs · 2026-01-26

Accept (Poster)